# INPP5E regulates CD3ζ enrichment at the immune synapse by phosphoinositide distribution control

Tzu-Yuan Chiu [1,7], Chien-Hui Lo[2], Yi-Hsuan Lin[2], Yun-Di Lai[2], Shan-Shan Lin[3], Ya-Tian Fang[1], Wei-Syun Huang[2], Shen-Yan Huang[4], Pei-Yuan Tsai[4], Fu-Hua Yang[1], Weng Man Chong[1], Yi-Chieh Wu[5], Hsing-Chen Tsai [5,6], Ya-Wen Liu [3], Chia-Lin Hsu[4], Jung-Chi Liao [1,8✉] & Won-Jing Wang [2✉]

The immune synapse, a highly organized structure formed at the interface between T lymphocytes and antigen-presenting cells (APCs), is essential for T cell activation and the adaptive immune response. It has been shown that this interface shares similarities with the primary cilium, a sensory organelle in eukaryotic cells, although the roles of ciliary proteins on the immune synapse remain elusive. Here, we find that inositol polyphosphate-5-phosphatase E (INPP5E), a cilium-enriched protein responsible for regulating phosphoinositide localization, is enriched at the immune synapse in Jurkat T-cells during superantigen-mediated conjugation or antibody-mediated crosslinking of TCR complexes, and forms a complex with CD3ζ, ZAP-70, and Lck. Silencing INPP5E in Jurkat T-cells impairs the polarized distribution of CD3ζ at the immune synapse and correlates with a failure of $PI(4,5)P_2$ clearance at the center of the synapse. Moreover, INPP5E silencing decreases proximal TCR signaling, including phosphorylation of CD3ζ and ZAP-70, and ultimately attenuates IL-2 secretion. Our results suggest that INPP5E is a new player in phosphoinositide manipulation at the synapse, controlling the TCR signaling cascade.

[1] Institute of Atomic and Molecular Sciences, Academia Sinica, Taipei 106319, Taiwan. [2] Institute of Biochemistry and Molecular Biology, National Yang Ming Chiao Tung University, Taipei 112304, Taiwan. [3] Institute of Molecular Medicine, National Taiwan University, Taipei 10002, Taiwan. [4] Institute of Microbiology and Immunology, National Yang Ming Chiao Tung University, Taipei 112304, Taiwan. [5] Graduate Institute of Toxicology, College of Medicine, National Taiwan University, Taipei 100233, Taiwan. [6] Department of Internal Medicine, National Taiwan University Hospital, Taipei 100233, Taiwan. [7] Present address: The Scripps Research Institute, La Jolla 92037, USA. [8] Present address: Syncell Inc., Taipei 115202, Taiwan. ✉email: jungchiliao@gmail.com; wangwj@nycu.edu.tw

T-cell activation plays a vital role in the cell-mediated adaptive immune response by engaging the T-cell receptor (TCR) with a specific antigen peptide/major histocompatibility complex (pMHC) ligand presented on antigen-presenting cells (APCs)[1]. The interaction between the TCR and the pMHC complex initiates the formation of a specialized structure known as the immune synapse[2]. The immune synapse is characterized by distinct subdomains, including the central supramolecular activating complex (cSMAC), peripheral SMAC (pSMAC), and distal SMAC (dSMAC). The cSMAC, located at the center of the synapse, is the site where the TCR and CD28 molecules accumulate upon conjugation. The concentration of receptors and signaling proteins in the cSMAC is crucial for efficient T-cell signaling and activation. Surrounding the cSMAC is the pSMAC, which contributes to the stabilization of the immune synapse. In addition to the pSMAC and cSMAC, there is an outer ring, the dSMAC, enriched with actin filaments and actin-associated proteins, regulating the cytoskeletal reorganization at the immune synapse. Various pathways for TCR recruitment to the immune synapse have been investigated, such as passive lateral diffusion within the plasma membrane[3,4], vesicular transport through recycling endosome trafficking[5], and cytoskeleton-mediated active movement[6]. All these routes are important in facilitating T-cell signaling.

Early events in the proximal signaling of TCR involve the Src family kinase Lck, which phosphorylates the immunoreceptor tyrosine-based activation motifs (ITAMs) of CD3[7,8]. The phosphorylated ITAMs serve as the docking sites for the recruitment of tyrosine kinase ZAP-70 (zeta-chain-associated protein kinase 70). This leads to downstream signaling cascades, including the activation of linker of activated T cells (LAT) and phospholipase Cγ1 (PLCγ1). Concurrently, T cells undergo rapid rearrangement of actin cytoskeletons and microtubules as the centrosome polarizes toward the immune synapse[9,10]. These processes ultimately lead to the formation of the mature immune synapse and the induction of functional effector responses such as cytokine secretion and exosome release[11,12].

The centrosome, which serves as the major microtubule-organizing center (MTOC) in animal cells, plays critical roles in microtubules organization and acts as the template for primary cilia formation. The similarities between the immune synapse and primary cilia have been increasingly recognized. First, the centrosome polarization at the immune synapse in cytotoxic T lymphocytes (CTL) and its resemblance to centriole docking during ciliogenesis lead to the immune synapse being referred to as a "frustrated cilium"[13,14]. It is known that docking the centriole to the membrane during ciliogenesis requires the presence of the centriole distal appendage structure at the centriole. Several proteins have been identified to locate at centriole distal appendages, including CEP164, SCLT1, CEP83, CEP89, FBF1, LRRC45, ANKRD26, and TTBK2[15–17]. Some of these proteins have also been found to localize at the immune synapse in CTL, and the loss of CEP83 impairs CTL secretion[18]. Moreover, both immune synapse and primary cilia formation involve actin reorganization and MTOC polarization to facilitate cellular events. The Bardet–Biedl syndrome complex (BBSome), consisting of eight subunits, cooperates with the IFT-B complex and participates in the trafficking of ciliary cargoes and MTOC-associated functions[19,20]. BBSome protein BBS1 also helps polarize the centrosome toward the immune synapse and clears F-actin localized around the immune synapse[21]. There are also surprising parallels in the trafficking machinery between primary cilia and immune synapses. Several intraflagellar transport proteins (IFTs), typically associated with primary cilia, are expressed in hematopoietic cells and are involved in T lymphocyte activation. For instance, IFT20 is known to recycle the TCR/CD3 complex and

recruit LAT to the immune synapse, supporting the signaling events occurring at the immune synapse[22–26]. Finally, both structures exhibit concentrated signal transduction activities. At the immune synapse, signaling molecules accumulate in specific regions like the cSMAC to facilitate efficient T-cell signaling. In the primary cilium, the enrichment of receptors in the ciliary membrane receives and transmits signals from the environment, enabling the primary cilium to function as a central signaling hub within the cell.

INPP5E is an enzyme responsible for converting $PI(4,5)P_2$ and $PIP_3$ into $PI(4)P$ and $PI(3,4)P_2$ by removing the 5-phosphate group from them[27]. In primary cilia, $PI(4,5)P_2$ is exclusively found in the ciliary base, while $PI(4)P$ is localized to the ciliary membrane. This specific distribution of phosphoinositides is crucial for Hedgehog signaling, as INPP5E depletion leads to the recruitment of Hedgehog signaling inhibitors like TULP3 and Gpr161[28,29]. Interestingly, the immune synapse undergoes changes in membrane composition during centrosome docking, similar to the primary cilium. When T-cell receptors engage with APCs, phospholipase C-gamma (PLCγ) is rapidly recruited to the immune synapse, leading to the breakdown of $PI(4,5)P_2$[30]. The clearance of $PI(4,5)P_2$ from the cSMAC is essential for efficient centrosome docking and subsequent granule secretion[31]. On the other hand, the phosphoinositides, especially $PI(4,5)P_2$ are reported to play important roles in establishing and maintaining the immune synapse[32]. It is known that the dynamic regulation of actin polymerization at the immune synapse facilitates the delivery of activating signals to the T cell that promotes the communication between the T cell and the APCs. Since $PI(4,5)P_2$ binds to a variety of actin-regulating proteins, it is likely that $PI(4,5)P_2$ coordinates the assembly and disassembly of actin filaments by binding to these actin-regulating proteins[33,34]. Considering the role of INPP5E in controlling the levels of $PI(4,5)P_2$ and the striking similarities between primary cilia and the immune synapse, it is possible that INPP5E might have a role in the immune synapse by regulating the distribution of phosphoinositides.

Here, we investigated whether additional ciliary proteins were present at the immune synapse. We observed that several ciliary proteins, including INPP5E, were recruited to the immune synapse. We also discovered that INPP5E interacted with CD3ζ, ZAP-70, and Lck during T-cell activation. This interaction played a crucial role in controlling CD3ζ recruitment toward the immune synapse, sustained proximal TCR signaling, facilitating NFκB nuclear translocation, and promoting the expression of IL-2. The effect of INPP5E in immune synapse formation occurred with $PI(4,5)P_2$ redistribution at the immune synapse, suggesting the potential phosphatase role of INPP5E on immune synapse formation. Our data reveal how INPP5E is co-opted as a regulator of early signaling and receptor recruitment during T-cell activation.

## Results

**Ciliary proteins are expressed in T cells**. We first tested whether any additional cilia-associated proteins localized close to the immune synapse during T-cell activation. Proteins are known to localize at the centriole distal end, the transition zone, and the cilium were stained in conjugates of Jurkat T-Raji B cells. Localization changes of these proteins were studied upon T-cell activation by means of staphylococcal enterotoxin E (SEE) engagement. As expected, without SEE, CD3ζ localized in the cytosol, presumably at the regions of the Golgi apparatus/endosomes, whereas with SEE, CD3ζ was recruited to the immune synapse. Centriole protein centrin 2 (CETN), ciliary suppression protein CEP97, and centriole distal appendage protein CEP164

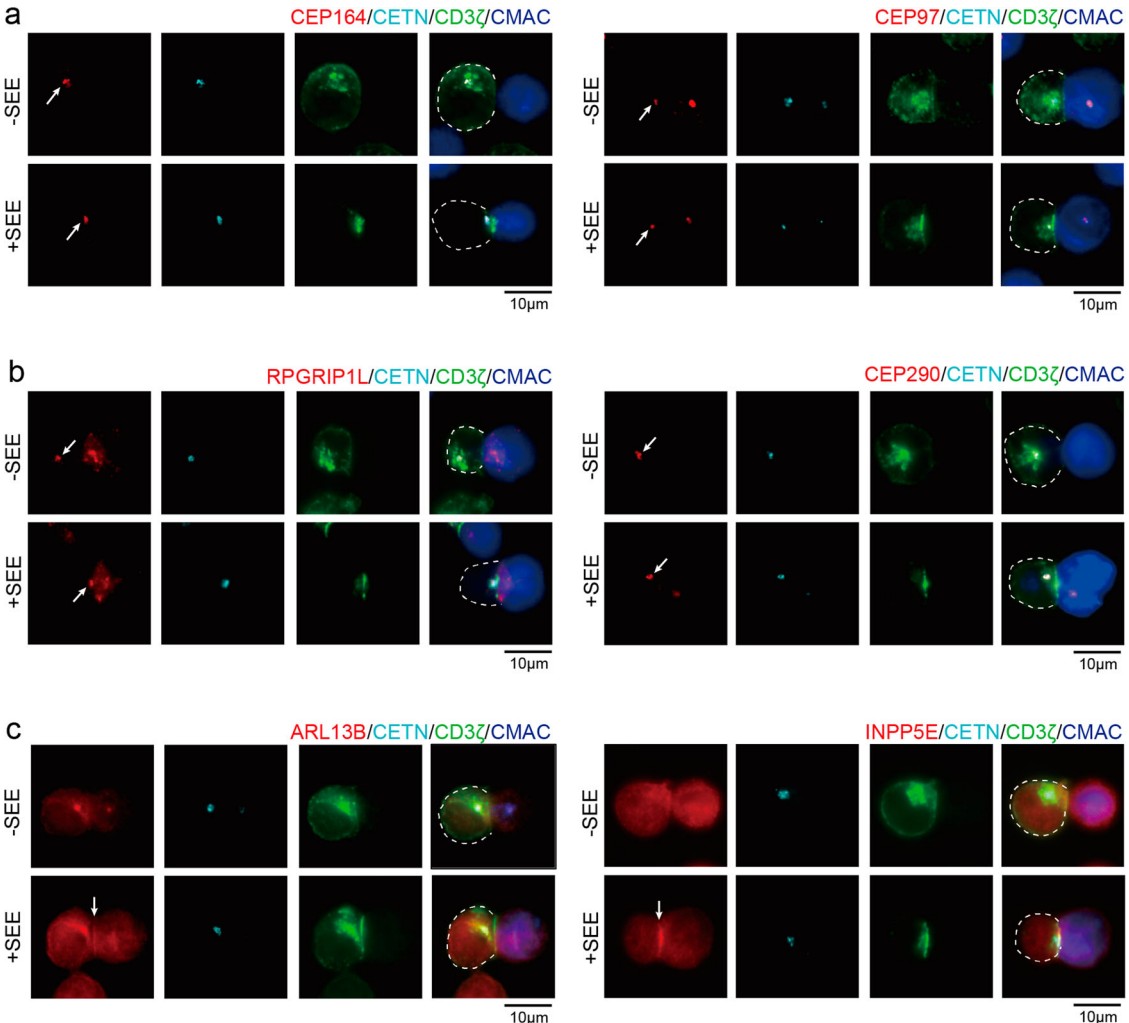

**Fig. 1 Localization of cilium-associated proteins in resting and activated T cells.** Immunostaining of **a** proteins at the centriole distal end (CEP97, CEP164), **b** transition zone proteins (RPGRIP1L, CEP290), and **c** cilium-enriched proteins (ARL13B, INPP5E) in conjugates of Jurkat T cells (Left) and CMAC-labeled Raji B cells (Right), in the presence (+) or absence (−) of SEE. Cells were co-stained with anti-centrin 2 (CETN, cyan) and anti-CD3ζ (green) antibodies as centriole and immune synapse markers. Arrows indicate the localization of each cilium-associated protein. Images are representative of at least three independent experiments. T cells are depicted in dotted lines. Scale bar: 10 μm.

were localized at centrioles, which are close to CD3ζ both in the absence and presence of SEE (Fig. 1a). In the presence of SEE, these proteins were polarized to the immune synapse[9,18]. We also examined the transition zone proteins, including RPGRIP1L and CEP290[35] (Fig. 1b). Interestingly, RPGRIP1L and CEP290 localized at the centrioles, where they were polarized to the cell contact sites during superantigen-mediated conjugation. That is, at least some of the transition zone proteins were present in Jurkat cells, implying possible involvement of the ciliary machinery in Jurkat cells.

We further examined ciliary proteins, including ARL13B and INPP5E, to see whether they were adjacent to the immune synapse. We found that ARL13B accumulated at the immune synapse when conjugated with SEE-loaded APCs, similar to the result reported by others[36] (Fig. 1c). For the phosphatase INPP5E, we found it localized with the centriole in the absence of SEE, similar to those found in ciliated cells (Supplementary Fig. 1a)[37]. We found that INPP5E was centered on MTOC at the interphase and was separated into two parts at the metaphase (Supplementary Fig. 1b). INPP5E formed a line pattern similar to that of CD3ζ at the interface of T cells and APCs when Jurkat cells were conjugated with SEE-loaded APCs. While INPP5E has been

considered as a Golgi-localized enzyme[32,38], the line pattern in Jurkat cells seems to be associated with membrane-anchored CD3ζ (Supplementary Fig. 1c). It is thus important to examine whether this cilium-specific phosphatase is indeed associated with immune synapse formation.

**INPP5E accumulates at the immune synapse in activated T cells.** To validate the localization of INPP5E at the immune synapse, we knocked down endogenous INPP5E by transfecting Jurkat cells with either the control (siCtrl) or INPP5E-specific siRNA (siINPP5E). A clear depletion of INPP5E signals at about 70-kDa was observed in INPP5E-knockdown (siINPP5E) cells (Fig. 2a). In resting Jurkat cells, the enrichment of INPP5E at the centrioles was not seen in siINPP5E cells (Supplementary Fig. 2a). Under SEE treatment, although non-specific binding of antibody was found in both T cells and APCs, the polarization of INPP5E toward the immune synapse was significantly reduced in siINPP5E cells (number of events in conjugation 38.09%, $n = 111$ versus control cells 71.52%, $n = 122$), including the use of the recruitment index with proteins at T-cell-APC contact sites[39] (Fig. 2b, c; Supplementary Fig. 2b). We then examined INPP5E

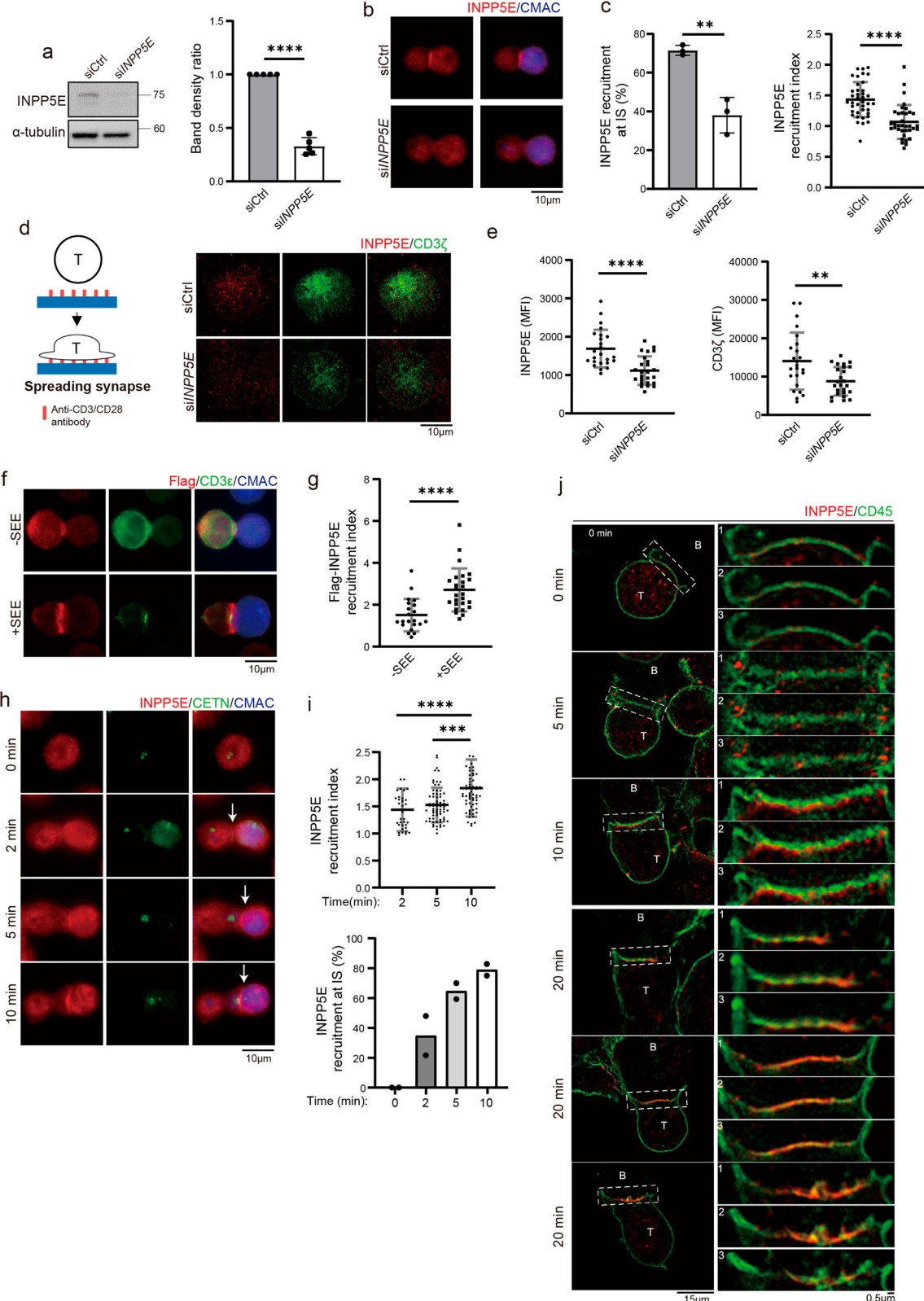

activated (Fig. 2d, e). We also transfected INPP5E with an N-terminal Flag tag into Jurkat cells and examined the Flag signals at the T-cell-APC contact site. Under SEE treatment, Flag-INPP5E was seen at the synapse (Fig. 2f, g), assuring the synapse localization of INPP5E in Jurkat T cells.

The spatial and temporal distribution of INPP5E during immune synapse formation was further examined. The

**Fig. 2 INPP5E accumulates at the immune synapse in activated T cells. a** Jurkat cells were transfected with either control (siCtrl) or *INPP5E*-specific siRNA (si*INPP5E*). INPP5E expression was analyzed by immunoblotting. Gel bands were quantified. Error bars indicate mean ± SD. $N = 5$ by student *T*-test. ****$P < 0.0001$. **b** Jurkat cells were transfected with either siCtrl or si*INPP5E*. SEE-coated Raji cells conjugated to Jurkat T cells were fixed and immunostained for INPP5E. Scale bar: 10 μm. **c** In the left panel, the percentage of INPP5E recruitment in conjugates at immune synapses from four independent experiments. $n = 122$ conjugates for siCtrl, and $n = 111$ conjugates for si*INPP5E*. In the right panel, the recruitment index of INPP5E was quantified. $n = 39$ for siCtrl and $n = 35$ for si*INPP5E*. Error bars indicate mean ± SD. Unpaired student *T*-test. **$P < 0.005$. ****$P < 0.0001$ **d** Jurkat cells were transfected with either control or si*INPP5E*. Representative TIRF images of INPP5E and CD3ζ in spreading Jurkat T cells on anti-CD3/CD28-coated coverslips for 10 min are shown. Scale bar: 5 μm. A schematic figure of the spreading assay is shown. **e** The mean fluorescence intensity (MFI) of INPP5E and CD3ζ from (**d**) was quantified. Data were obtained from three independent experiments. $n = 24$ cells for siCtrl, and $n = 25$ cells for si*INPP5E*. Unpaired student T-test. **$P < 0.005$. ****$P < 0.0001$. **f** Flag-INPP5E was expressed in conjugates of Jurkat T cells and CMAC-labeled SEE-pulsed APCs, followed by immunostaining. Cells were co-stained with an anti-CD3ε antibody as the immune synapse marker. Scale bar: 10 μm. **g** The recruitment of INPP5E at immune synapses from (**f**) was quantified. $N = 4$. $n = 21$ conjugates for -SEE, and n = 26 conjugates for +SEE. Unpaired student *T*-test. ****$P < 0.0001$. **h** Immunostaining of INPP5E in T-cell-APC conjugates with different time points. Scale bar: 10 μm. **i** Quantification of the INPP5E recruitment index (upper panel) and the INPP5E enrichment events (lower panel) in conjugates were from two independent experiments. $n = 34$ for 2 min, $n = 70$ for 5 min, and n = 63 for 10 min. 0 min represent an experiment where cells were fixed before adding APCs. Error bars indicate mean ± SD. One-way ANOVA analysis. *$P \leq 0.05$. **$P \leq 0.005$. ***$P < 0.001$. ****$P < 0.0001$. **j** Three-dimensional structured illumination microscopy (3D-SIM) images of INPP5E in T-cell APC conjugating at different time points. CD45 was used as a T-cell surface marker. Enlarged figures are shown on the right. The numbers 1-3 indicate z-stack intervals from the top of each cropped 3D-SIM image. Scale bars are as indicated.

immunostaining revealed that the INPP5E signal appeared faintly at the immune synapse 2 min after conjugation and increased 5–10 min after conjugation (Fig. 2h, i). Three-dimensional structured illumination microscopy (3D-SIM) was then performed to visualize INPP5E signals at the immune synapse in detail. By using CD45 as the cell surface marker, INPP5E was observed to localize at the synapse with dotted puncta 5 min after T-cell-APC conjugation and became a line pattern 10 min after conjugation (Fig. 2j). A short distance between cell surface marker CD45 and INPP5E could still be observed at 10 min, while INPP5E moved toward and localized with CD45 at 20 min (19 out of 25 conjugates). To determine whether the recruitment of INPP5E at the immune synapse was not specific to Jurkat cells, we analyzed the expression of INPP5E between Jurkat cells and primary T cells from healthy donors. Our results showed that INPP5E expression levels were comparable between the two cell types (Supplementary Fig. 2c). Moreover, we purified mouse primary T cells and used anti-CD3/CD28-coated beads to induce immune synapse formation. After 20 min of conjugation, INPP5E signals were detected at the sites of T-cell-bead conjugation (Supplementary Fig. 2d, e). Together, these results confirmed that INPP5E was recruited and enriched at the immune synapse during T-cell activation and is not limited to specific cell types.

**The proline-rich domain is required for efficient INPP5E recruitment at the immune synapse.** To map the minimal domain required for INPP5E localization at the synapse, we reconstituted different truncations of Flag-tagged INPP5E, which have been described in previous studies[41,42], into Jurkat cells and observed their localization upon T-cell-APC conjugation (Fig. 3a, b). We found that the N-terminal portions of INPP5E (amino acids 1–289, 1–604, and 1–626) showed synaptic localization. However, the C-terminal INPP5E (amino acids 79–644 and 288–644) showed impaired synaptic recruitment (Fig. 3c, d). These results indicated that the synaptic recruitment domain of INPP5E was around the N-terminal region, i.e., the proline-rich domain (PRD), instead of the ciliary targeting sequence (CTS) of INPP5E known for primary cilia between the inositol polyphosphate phosphatase catalytic domain (IPPc) and the CaaX motif (Fig. 3e). Taken together, our results suggest that the proline-rich domain is required for efficient synaptic recruitment of INPP5E.

**INPP5E is required for CD3ζ recruitment to the immune synapse.** The TCR complex is known to be recruited to the

immune synapse when stimulating T cells either through APCs pulsed with peptide antigens or anti-CD3/CD28 treatment[43,44]. The clustering of the TCR/CD3 complex triggers activities of kinases and scaffold proteins to maintain the proximal TCR signaling[45–47] and further control T-cell activation[48]. As INPP5E was enriched at the immune synapse during T-cell activation, we examined whether INPP5E at the immune synapse played a role in regulating proximal TCR signaling. We capped the TCR/CD3 complex using anti-CD3 and anti-CD28 monoclonal antibodies, a procedure leading to TCR/CD3 clustering[49]. The immunoprecipitation assay revealed that INPP5E interacted with the complex CD3ζ, ZAP-70, and Lck in response to TCR engagement in a time-dependent manner (Fig. 4a)[48,50,51]. To test whether INPP5E directly interacted with these proteins, we used a heterologous HEK293T cell system to approach this question. We transfected CD3ζ-GFP, Lck, ZAP-70-Myc, and Flag-INPP5E together into HEK293T cells since full phosphorylation of CD3ζ requires Lck and ZAP-70 co-expression in COS-7 or HEK293T cells[52,53]. CD3ζ-GFP and ZAP-70-Myc were indeed phosphorylated when co-transfecting Lck in HEK293T cells (Supplementary Fig. 3a). Our results showed that CD3ζ, Lck, and ZAP-70 were detected in INPP5E immunoprecipitate (Fig. 4b). By co-expressing INPP5E and either CD3ζ, ZAP-70, or Lck separately in 293T cells, INPP5E was able to interact with these proteins individually (Fig. 4c). These results suggested that INPP5E interacted with the proximal TCR signaling proteins, and the phosphorylation of CD3ζ and ZAP-70 might not be required for their interaction with INPP5E. Notably, the portion of INPP5E responsible for interacting with CD3ζ was the C-terminal region (amino acids 288–644). Since the N-terminal INPP5E (amino acids 1–289) could not pull down CD3ζ-GFP, which indicated INPP5E might utilize different domains for immune synapse docking and CD3ζ interaction (Supplementary Fig. 3b).

To check whether INPP5E regulated the recruitment of CD3ζ and Lck to the immune synapse during T-cell activation, we examined the localization of endogenous CD3ζ and Lck in control and si*INPP5E* Jurkat cells that were conjugated with SEE-pulsed APCs. Our results showed that CD3ζ was recruited to the immune synapse in control cells, while INPP5E depletion decreased its accumulation (Fig. 4d, e). On the contrary, INPP5E depletion did not affect the recruitment of Lck at the immune synapse (Supplementary Fig. 3c), suggesting a different recruiting relationship between CD3ζ and Lck. Accumulation of CD3ζ at the pseudo-synapse was also significantly decreased in si*INPP5E* cells when Jurkat cells were activated with plate-bound anti-CD3/CD28 (Fig. 2d, e).

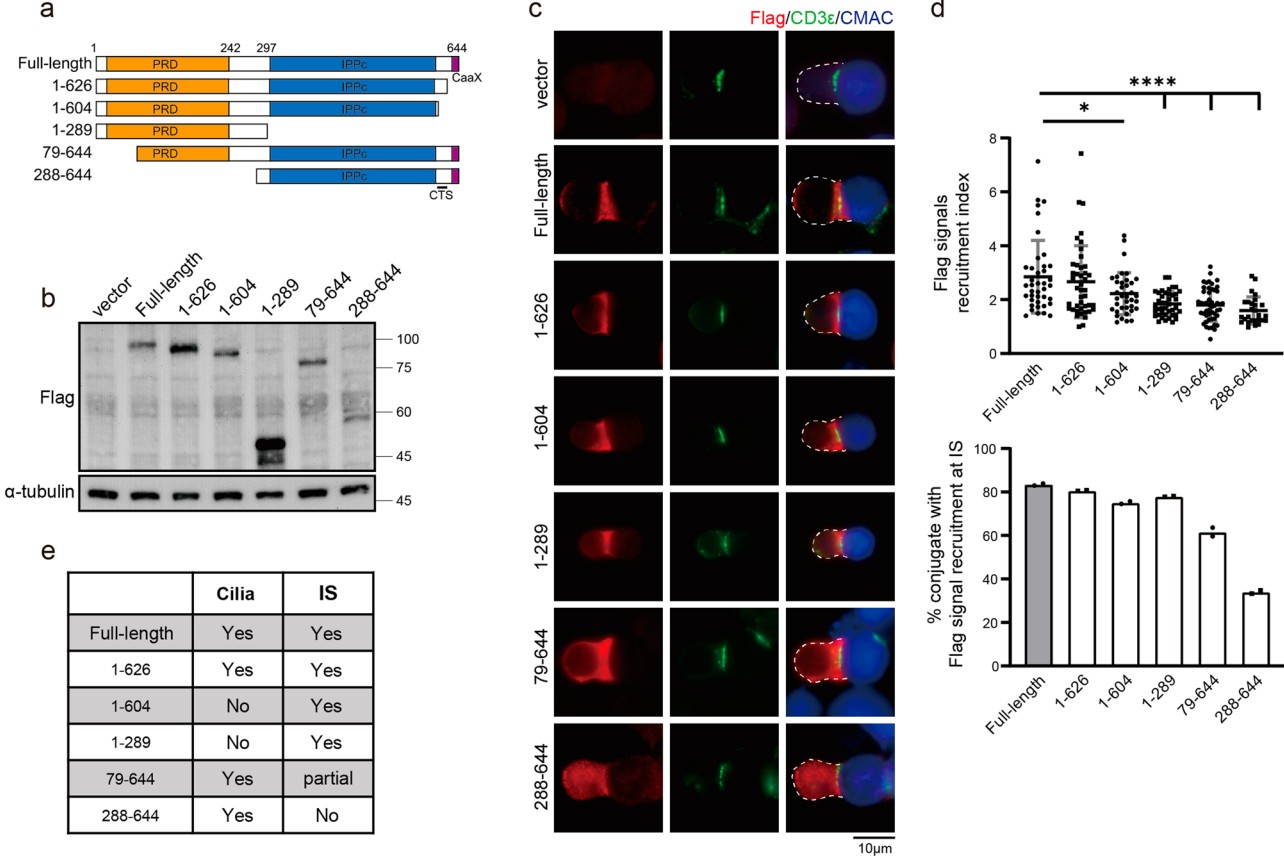

**Fig. 3 The proline-rich domain is required for efficient INPP5E recruitment at the immune synapse. a** Diagrams of INPP5E truncations. Proline-rich domain (PRD), inositol polyphosphate phosphatase catalytic (IPPc) domain, CaaX motif are marked in yellow, blue, and purple, respectively. The CTS indicates the ciliary targeting sequence. **b** Jurkat cells were transfected with different truncations of Flag-INPP5E. The expression levels of Flag-INPP5E truncations were examined by western blots. α-tubulin was used as the loading control. **c** Immunostaining of Flag-INPP5E truncations in conjugates of Jurkat T cells and CMAC-labeled SEE-pulsed APCs. Cells were co-stained with anti-CD3ε as an immune synapse marker. Scale bar: 10 μm. **d** Quantification of the Flag-INPP5E recruitment index (upper) and the percentage of conjugates with Flag-INPP5E truncations at the immune synapse (lower) is shown. The recruitment of Flag-INPP5E truncations to the immune synapse was shown by the scatter plot graph. 30–50 conjugates were quantified for each INPP5E mutant. Images are representative of two experiments. Error bars indicate mean ± SD. One-way ANOVA analysis. *$P \leq 0.05$. **$P \leq 0.005$. ****$P < 0.0001$. **e** Summary of INPP5E truncations in the localization of primary cilia and immune synapse (IS).

To further examine how INPP5E regulated spatial and temporal localization of CD3ζ at the immune synapse during T-cell activation, we ectopically expressed CD3ζ fused with EGFP into control and si*INPP5E* cells[54,55]. We co-cultured Jurkat cells with APCs before fixation, and the conjugates with depletion of filamentous actin (F-actin) at the center of the synapses were counted. In the absence of SEE, CD3ζ-GFP was evenly distributed on the plasma membrane (Fig. 4f, upper). In the presence of SEE, CD3ζ-GFP mostly accumulated at the immune synapse in control cells, whereas clustering of CD3ζ-GFP was considerably impaired in si*INPP5E* cells (recruitment index 2.17, $n = 41$ versus control cells 4.53, $n = 44$) (Fig. 4f, middle and lower; and Fig. 4g). Live imaging of CD3ζ-GFP showed that CD3ζ-GFP started to accumulate at the immune synapse 1.5 min after conjugation in control cells, while clustering of CD3ζ-GFP at the T-cell-APC contact site was impaired in si*INPP5E* cells (Fig. 4h; *en face* images at Supplementary Fig. 3d). Collectively, these data indicate that INPP5E is required for CD3ζ clustering at the immune synapse during T-cell activation.

**INPP5E modulates PI(4,5)P₂ and CD3ζ distribution at the immune synapse.** One factor known to regulate TCR/CD3 recruitment is phosphoinositide[56–58]. In primary cilia, INPP5E hydrolyzes PI(4,5)P2 to PI4P to regulate the localization of ciliary

Hedgehog signaling inhibitors[29,59,60]. Dysregulation of phosphoinositide in INPP5E-deficient mice results in the lethality of embryos and ciliopathies in humans[59,61]. A panel of phosphoinositide probes has been identified to study the distribution of phosphoinositides (Supplementary Fig. 4a). To test whether INPP5E controls local levels of its substrate at the immune synapse, we expressed the pleckstrin homology (PH) domain of PLCδ fused to GFP (PH-PLCδ) in control and si*INPP5E* cells to monitor PI(4,5)P2 level[62,63]. We allowed T-B conjugates to form before fixation, and the 3D structure at the synapse was reconstituted from stacked confocal microscopy images. Cells treated with control siRNA showed negligible PH-PLCδ (Fig. 5a, upper) fluorescence at the center of the T-cell-APC site, indicative of limited PI(4,5)P2 molecules at the inner part of the immune synapse. In contrast, the distribution of PH-PLCδ was relatively uniform across the entire contact interface of the immune synapse in si*INPP5E* cells (clearance percentage 46.1% versus control cells 67.8%) (Fig. 5a, lower; and Fig. 5b). Dynamics of PH-PLCδ at the immune synapse was also monitored using live-cell imaging in control and si*INPP5E* cells. In control cells, PI(4,5)P2 level at the center of the T-cell-APC site was sharply reduced 3 min after conjugation, and complete clearance was observed at 5 min. In contrast, the distribution of PI(4,5)P2 at the synapse was not changed in si*INPP5E* cells, suggesting that INPP5E

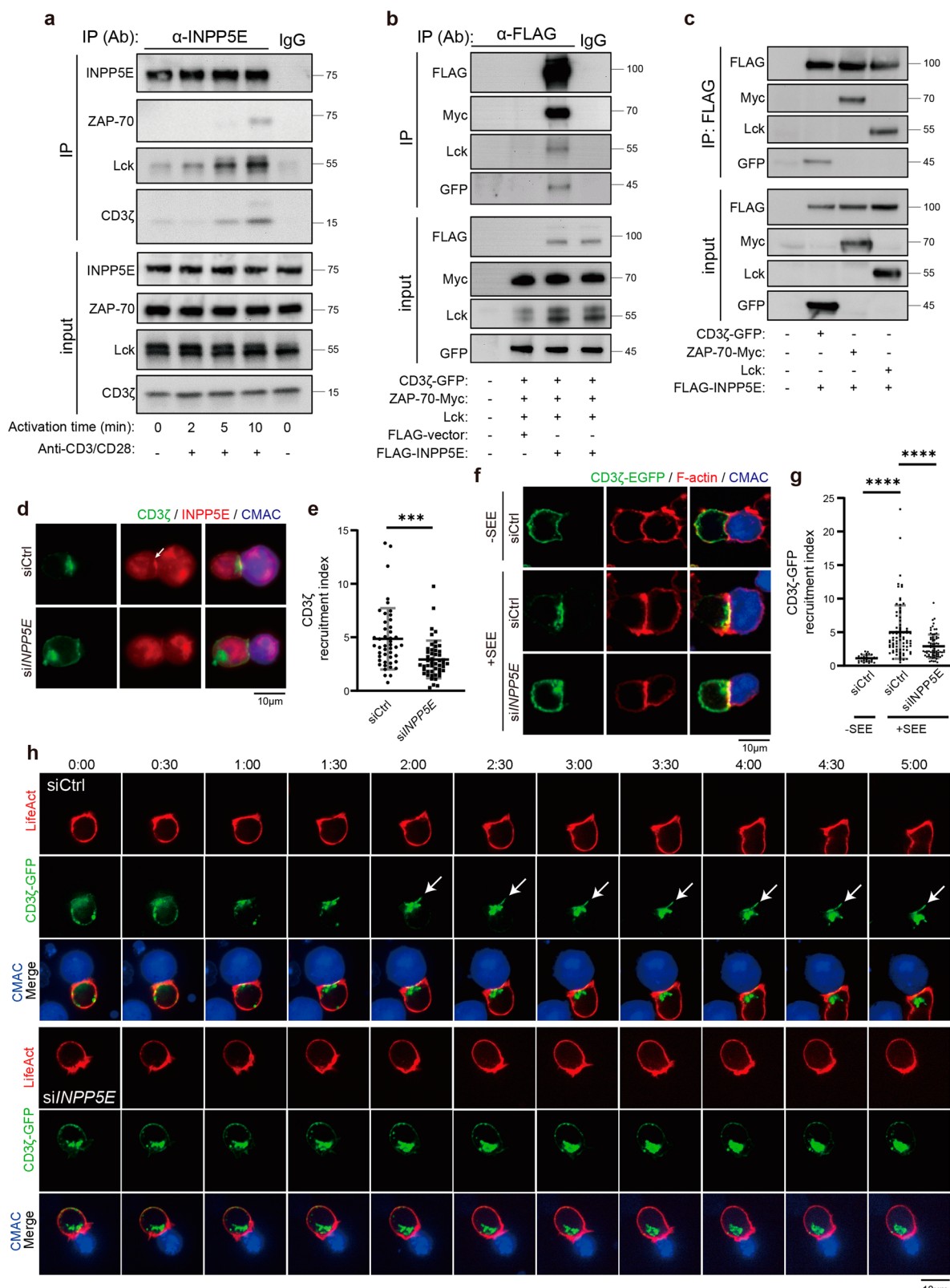

regulated PI(4,5)P2 dynamics at the synapse site (Fig. 5c, d). That is, the clearance of PI(4,5)P2 at the inner part of the immune synapse was prevented in si*INPP5E* cells, suggesting that INPP5E controlled the PI(4,5)P2 conversion at the synapse. We also confirmed this observation by transfecting Tubby-GFP, another PI(4,5)P2 sensor[31,32]. In control cells, Tubby-GFP signals were again abolished at the synapse; on the other hand, PI(4,5)P2

clearance was impaired in si*INPP5E* cells, suggesting that INPP5E regulated the PI(4,5)P2 level at the synapse site (Supplementary Fig. 4b, c).

To assess more directly whether modulating PI(4,5)P2 distribution affected CD3ζ-GFP recruitment, we used phosphatidylinositol-4-phosphate 5-kinase type 1 gamma (PIP-KIγ) fused to mCherry to increase the PI(4,5)P2 population on

**Fig. 4 INPP5E is required for CD3ζ recruitment to the immune synapse. a** Jurkat T cells were activated by crosslinking with the anti-CD3/CD28 antibodies for 2–10 min. The total cell lysates from each time point were immunoprecipitated (IP) with anti-INPP5E antibody. The immunoprecipitates were resolved by 12% SDS-PAGE followed by immunoblotting with indicated antibodies. Representative images are shown. IgG lane was used here as the control. **b, c** HEK293T cells were transfected with indicated plasmids. Cells were lysed and immunoprecipitated (IP) with the anti-Flag antibody. The immunoprecipitates were resolved by 10% SDS-PAGE followed by western blot with indicated antibodies. **d** Jurkat cells were transiently transfected with either siCtrl or si*INPP5E*. Immunostaining of CD3ζ and INPP5E in conjugates of Jurkat T cells and CMAC-labeled SEE-pulsed APCs. Scale bar: 10 μm. **e** Quantification of CD3ζ recruitment at immune synapse from (**c**). n = 46 conjugates for the siCtrl, and n = 49 conjugates for si*INPP5E*. Scale bar = 10 μm. Error bars indicate mean ± SD. Unpaired student T-test. ***P < 0.001. Arrows indicate localization of INPP5E. **f** Jurkat cells were transfected CD3ζ-GFP together with either siCtrl or si*INPP5E*. Immunostaining of CD3ζ-GFP in conjugates of Jurkat T cells and CMAC-labeled APCs, in the absence (−, upper) or presence (+, middle) of SEE. Cells were conjugated for 10 min, and co-stained with phalloidin to visualize F-actin. Upper and middle, control siRNA transfected cells; lower, si*INPP5E* transfected cells. Scale bar = 10 μm. **g** The recruitment index of CD3ζ-GFP at the immune synapse was quantified. Dotted black squares indicate the regions that were selected for quantifying the GFP fluorescence intensity. N = 4. n = 30 conjugates for siCtrl -SEE, n = 81 conjugates for siCtrl +SEE, and n = 73 conjugates for si*INPP5E*. Error bars indicate mean ± SD. One-way ANOVA analysis. ****P < 0.0001. **h** Jurkat cells were transfected with either siCtrl or si*INPP5E* at day 1, while CD3ζ-GFP and LifeAct-TagRFP are transfected at day 3. Cells were analyzed at day 4. Live images of cells conjugates were recorded for 5 min. Arrows indicate the localization of CD3ζ-GFP at the T-cell-APC contact site. The represented images are shown. The results were from 3–5 cells for each condition from two independent experiments.

the plasma membrane[32]. mCherry-PIPKIγ was first introduced into parental Jurkat cells, followed by conjugating cells with SEE-loaded APCs. We found that signals of mCherry-PIPKIγ, which resembled PI(4,5)P2 levels, were evenly distributed at the synapse (Supplementary Fig. 4b, c). Next, we co-transfected mCherry-PIPKIγ and CD3ζ-GFP into Jurkat cells to visualize CD3ζ distribution when the clearance of PI(4,5)P₂ at the SMAC was impaired. Excitingly, recruitment of CD3ζ-GFP was largely diminished in the PIPKIγ-expressed cells, while the clustering of GFP signals remained accumulated at the immune synapse in mCherry control cells (Fig. 5e, f). This result indicated that dysfunction of immune synapse response in si*INPP5E* cells could be caused by abnormal PI(4,5)P₂ levels and/or distribution, similar to the findings in primary cilia[37].

It is known that the level of PI(4,5)P2 on the membrane is also regulated by the activity of several phosphoinositide-directed kinases and phosphatases. Here we examined the influence of INPP5E on the recruitment of known PI(4,5)P2 regulators at the immune synapse. The control and si*INPP5E* Jurkat cells were conjugated with Raji cells and stained with antibodies specific for known PI(4,5)P2 regulators, such as PIK3R1 and PIP5K1A. PIK3R1 is the p85 subunit of phosphoinositide 3-kinase, whereas PIP5K1A is a PI4P kinase that converts PI4P into PI(4,5)P₂. Our results showed that INPP5E depletion did not affect the recruitment of PIK3R1 or PIP5K1A at the immune synapse (Supplementary Fig. 4d). We also checked PI(4,5)P₂ lipase PLCγ1, which converts PI(4,5)P₂ into inositol 1,4,5-triphosphate and diacylglycerol. Our results showed that knockdown of INPP5E decreased PLCγ1 recruitment at the synapse (Fig. 5g, h). Given that F-actin clearance is impaired during PI(4,5)P₂ inhibition, we examined the potential influence of INPP5E in regulating F-actin clearance at the immune synapse[31,32]. In Jurkat-APC conjugation experiments, the F-actin staining showed that F-actin clearance at immune synapse was impaired in si*INPP5E* cells, as evidenced by a clearance percentage of 53.6 compared to 73.6% in control cells (Supplementary Fig. 4e). Meanwhile, we did not observe significant alterations in INPP5E-depleted cells when quantifying the spreading areas of F-actin on anti-CD3/CD28-coated slides (Supplementary Fig. 4f). Thus, we propose that INPP5E directly modulates PI(4,5)P2 level at the synapse, which in turn affects the recruitment of CD3ζ and F-actin clearance.

**INPP5E is required for efficient proximal TCR signaling and effector functions.** TCR recruitment toward the immune synapse is an outcome of T-cell activation[26,64]. Moreover, dephosphorylation of PI(4,5)P2 by exogenous phosphatase Inp54p in

mouse T-cell lymphoma enhances the lateral mobility of the TCR/CD3 complex on the plasma membrane, thus increasing TCR activation and signaling[58]. To validate whether the recruitment of INPP5E at immune synapses was important for proximal TCR signaling, control and si*INPP5E* Jurkat cells were activated with anti-CD3/CD28 for various durations to stimulate the TCR response[65,66]. At 5 and 15 min, si*INPP5E* cells showed a considerable attenuation of both CD3ζ and ZAP-70 phosphorylation levels upon T-cell activation as compared to the response in control cells, while the total amounts of CD3ζ and ZAP-70 were not affected (Fig. 6a; quantification in Supplementary Fig. 5a). PLCγ1 phosphorylation was also slightly decreased, though it was statistically insignificant. The active Lck (Lck Y394) level remained similar between control and si*INPP5E* cells. Furthermore, control and si*INPP5E* cells were conjugated with SEE-loaded B cells for 10 min, and cells were fixed and stained with anti-phospho-CD3ζ (Y83) and phospho-ZAP-70 (Y493) antibodies. The MFI of phosphorylated proteins at the synapse was quantified. We found a significant decrease of CD3ζ and ZAP-70 phosphorylation signals in si*INPP5E* cells, suggesting that INPP5E sustained the initial activation of T cells (Fig. 6b, c).

To further examine the influence of INPP5E in early T-cell signaling events, we analyzed NFκB activity by monitoring its nuclear translocation[8]. The control and si*INPP5E* Jurkat cells were activated with anti-CD3/CD28 to stimulate the TCR response. The si*INPP5E* cells showed reduced NFκB nuclear translocation compared to control cells, suggesting the importance of INPP5E in early T-cell signaling events (Fig. 6d). CD40L, which is expressed on the surface of activated T cells, is the protein involved in the coordination of immune responses[67,68]. In comparison to the control cells, the percentage of CD40L-expressing cells was significantly reduced in si*INPP5E* Jurkat cells, further demonstrating the importance of INPP5E in coordinating immune responses (Fig. 6e). To determine the impact of INPP5E depletion on T-cell effector functions, we measured the surface expression of CD25 and the secretion amount of IL-2 in control and si*INPP5E* Jurkat cells after T-cell activation. While INPP5E knockdown only slightly reduced CD25 expression of, IL-2 secretion was significantly attenuated in si*INPP5E* cells compared to control cells, regardless of whether the cells were activated with superantigen-loaded Raji cells or anti-CD3/CD28 (Fig. 6f, g; Supplementary Fig. 5b)[69]. Finally, we confirm the importance of INPP5E at the immune synapse in regulating T-cell activation by measuring the mRNA level of *Il-2* in shCtrl and sh*Inpp5e*-transduced mouse pan T cells. The *Il-2* mRNA level in sh*Inpp5e*-transduced pan T cells was significantly reduced compared to shCtrl-transduced pan T cells, confirming the essential role of

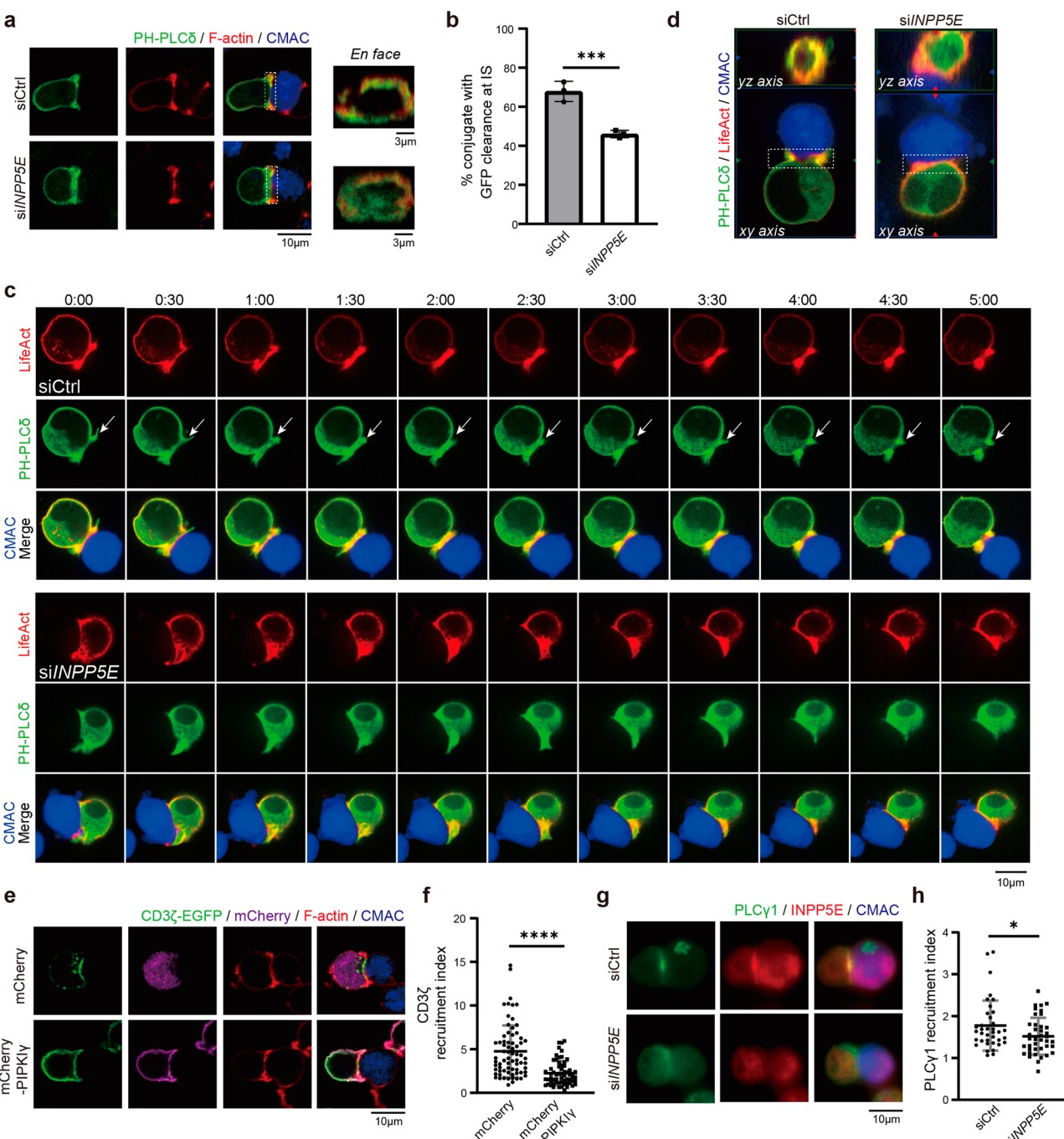

**Fig. 5 INPP5E modulates PI(4,5)P₂ environment at the immune synapse. a** Immunostaining was performed in PH-PLCδ-GFP transfected Jurkat T cells in conjugates. Images show in the xy plane (scale bar, 10 μm) or 1.0 μm 3D-reconstructed en face at the T-cell-APC contact site in the xz plane (scale bar, 3 μm). Cells were conjugated for 10 min and stained with an anti-GFP antibody. Phalloidin staining was used to visualize F-actin. Dotted white squares indicate the regions that were selected for 3D reconstruction. **b** Quantification of PH-PLCδ-GFP clearance at the immune synapse in conjugates. $n = 54$ conjugates for siCtrl, and $n = 74$ conjugates for si*INPP5E* cells. Unpaired *t*-test. **$P < 0.01$. **c** Jurkat cells were transfected with either siCtrl or si*INPP5E* at day 1, while PH-PLCδ-GFP and LifeAct-TagRFP were transfected at day 3. Cells were analyzed at day 4. Live images of cell conjugates were recorded for 5 min. Arrows indicate localization of PH-PLCδ-GFP at the T-cell-APC contact site. The representative images were shown. 3–5 cells were recorded for each condition. The data were obtained from two independent experiments. Arrows indicate PH-PLCδ at IS sites. **d** The 3D-reconstructed en face at immune synapse in the yz plane after 5 min conjugation. The dotted white squares indicate the regions that were selected for 3D reconstruction. **e** Jurkat cells were transfected with expression vectors encoding mCherry and mCherry-PIPKIγ. Immunostaining of CD3ζ-GFP in conjugates of Jurkat T cells and CMAC-labeled APCs, in the presence of SEE. Phalloidin staining was used to visualize F-actin. Upper, mCherry vector transfected cells; lower, mCherry-PIPKIγ transfected cells. Scale bar = 10 μm. **f** The CD3ζ-GFP recruitment index was quantified. $N = 4$. $n = 71$ conjugates for mCherry vector transfected cells, $n = 62$ conjugates for mCherry-PIPKIγ transfected cells. Error bars indicate mean ± SD. Unpaired t-test. ****$P < 0.0001$. **g** Jurkat cells were transfected with either siCtrl or si*INPP5E*. Immunostaining of PLCγ1 and INPP5E were performed in conjugates of Jurkat T cells and CMAC-labeled SEE-pulsed APCs. Arrows indicate localization of INPP5E. **h** The PLCγ1 recruitment index was quantified in conjugates. $N = 2$. $n = 37$ conjugates for siCtrl, and $n = 40$ conjugates for si*INPP5E*. Scale bar = 10 μm. Error bars indicate mean ± SD. Unpaired student T-test. *$P < 0.05$.

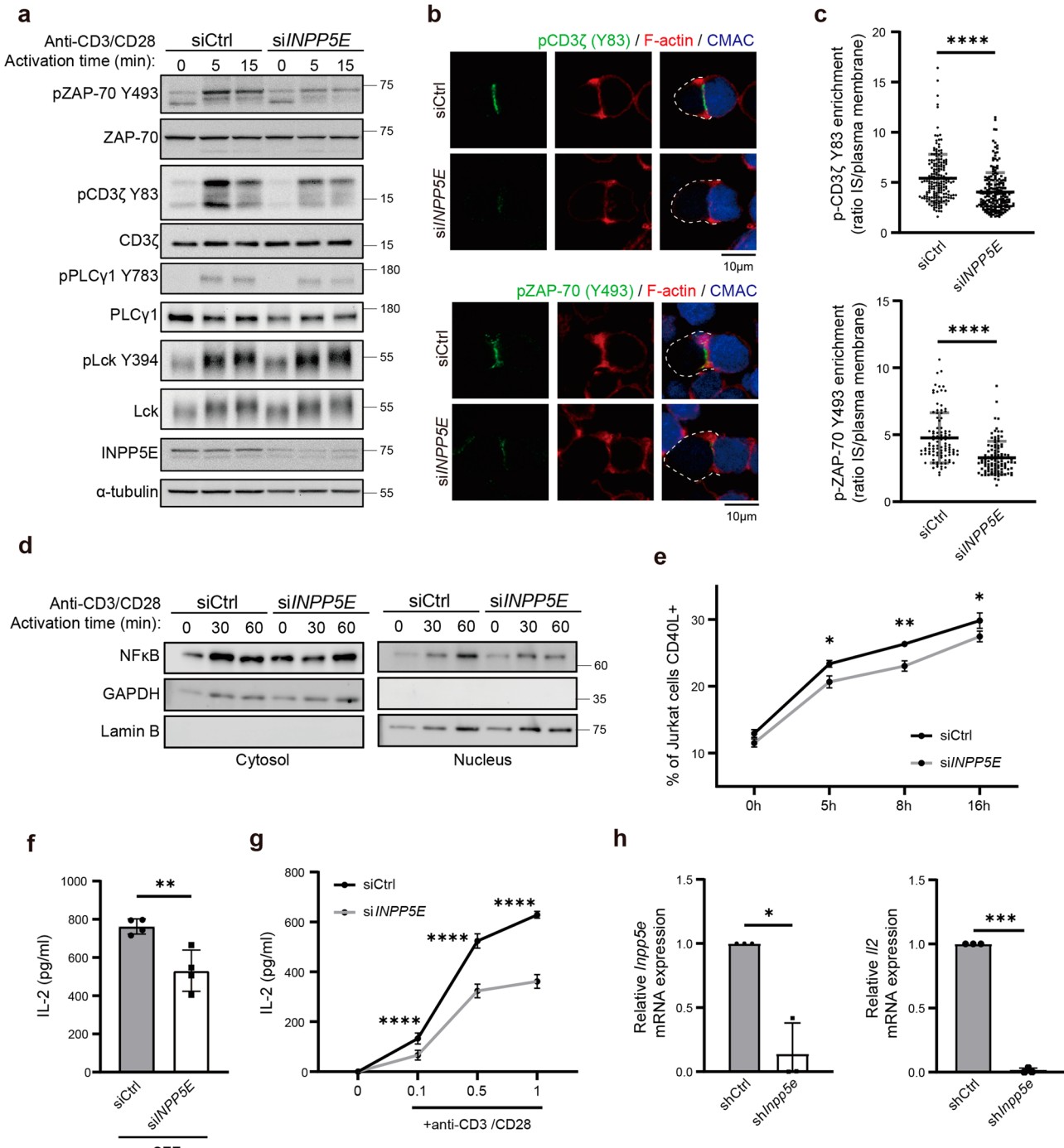

INPP5E in T-cell activation (Fig. 6h). Together, these data indicate that INPP5E is required for efficient T-cell activation through regulation of CD3ζ and ZAP-70 phosphorylation.

## Discussion
Recent studies have demonstrated that some proteins playing roles in the primary cilium are also recruited to the immune synapse[70,71], controlling TCR signaling and T-cell activation. Here we show that cilium-localized phosphatase INPP5E is indeed enriched at the immune synapse, where INPP5E interacts and regulates CD3 recruitment to the immune synapse by regulating the distribution of PI(4,5)P2 at the SMAC. Moreover, this regulation can affect the phosphorylation of CD3ζ and ZAP-70

and further enhance IL-2 secretion. Therefore, our results discover that INPP5E is a player in the immune synapse, mediating functions by its interplay with TCR complexes.

The TCR/CD3 complex is a crucial receptor complex that is involved in T-cell activation, and phosphoinositides are important in regulating the dynamics of this complex[58]. Studies have shown that dephosphorylation of PI(4,5)P2 through overexpressing the plasma membrane-localized inositol polyphosphate-5-phosphatase (Inp54p) in the mouse T-cell hybridoma causes the CD3ε cytoplasmic domain unbinding from the plasma membrane, resulting in a modest increase in TCR diffusion rate. In contrast, primary T cells that overexpress PIP5K, an enzyme involved in PI(4,5)P2 synthesis, increase the amount of PI(4,5)P2 at the immune synapse, leading to T-cell rigidity. It thus delays the recruitment of

**Fig. 6 INPP5E sustains efficient early proximal TCR signaling and the effector function. a** Jurkat cells were transfected with either siCtrl or si*INPP5E*. Immunoblots were performed to analyze the phosphorylation levels of indicated proteins in Ctrl or INPP5E-knockdown Jurkat T cells activated by crosslinking with anti-CD3/CD28 for the indicated times. α-tubulin and non-phosphorylated molecules were blotted as loading controls. Representative images are shown. **b** Jurkat cells were transfected with either siCtrl or si*INPP5E*. Immunostaining of pCD3ζ-Y83 (upper) and pZAP-70-Y493 (lower) in conjugates of the control or INPP5E-knockdown Jurkat T cells and CMAC-labeled SEE-pulsed APCs. Phalloidin staining was performed to visualize F-actin. Representative images are shown. **c** The enrichment index of phosphorylated proteins was quantified in conjugates. $N = 3$. $n = 183$ conjugates for the siCtrl, and $n = 199$ conjugates for si*INPP5E* in pCD3ζ experiments. $n = 101$ conjugates for Ctrl, and $n = 103$ conjugates for si*INPP5E* in pZAP-70 experiments. Error bars indicate mean ± SD. Unpaired Student $t$-test. ****$P < 0.0001$. **d** Jurkat cells were transfected with either control or *INPP5E*-specific siRNA followed by anti-CD3/CD28 incubation. Cells were subjected to nuclear and cytosolic fractionation followed by WB analysis with antibodies as indicated. Lamin B and GAPDH was used as nuclear and cytosolic marker, respectively. **e** Jurkat cells were transfected with either control or *INPP5E*-specific siRNA followed by anti-CD3/CD28 incubation. The percentage of CD40L-expressing cells was quantified. Error bars indicate mean ± SD. Paired student $t$-test. $P < 0.05$, **$P \leq 0.005$. **f, g** Jurkat cells were transfected with either control or *INPP5E*-specific siRNA. An ELISA assay showing the amount of IL-2 in the cultured supernatants in siCtrl or si*INPP5E* Jurkat T cells. In **f**, T cells were activated by SEE-pulsed APCs for 24 h. In **g**, T cells were activated by anti-CD3/CD28 for 24 h. The results were an average of at least three independent assays. Each assay has three technical replicates. *$P < 0.05$. **$P \leq 0.005$. **h** Mouse primary pan T cells were isolated from C57bl/6 mice and transduced with shCtrl or sh*Inpp5e* retrovirus. GFP positive cells were sorted and used to measure the *Il-2* mRNA level on day 4. The relative mRNA levels of *Inpp5e* and *Il-2* were quantified from three independent experiments. Error bars indicate mean ± SD. Paired student $t$-test. *$P < 0.05$, ***$P < 0.001$.

TCR complex and impairment of proximal TCR signaling[72]. The conclusion of these studies is consistent with the results in our works. We observe higher levels of PI(4,5)P2 at the center of the synapse in si*INPP5E* Jurkat cells, along with reduced recruitment of reconstituted CD3ζ-GFP to the immune synapse. This suggests that INPP5E may directly regulate PI(4,5)P$_2$ levels through its enzymatic activity or indirectly by interacting with other signaling molecules, affecting the diffusion of CD3ζ at the plasma membrane. Another possibility is that INPP5E regulates vesicular transport, which delivers signaling proteins like CD3ζ, to the immune synapse during T-cell activation[5]. This hypothesis finds supports in published studies, which suggest that the delivery of LAT and CD3ζ to the immune synapse happens through a mechanism independent of the one that transports Lck[55]. Additionally, the intraflagellar transport protein IFT20 has been shown to regulate CD3/LAT trafficking and is crucial for CD3 accumulation in the central cSMAC[24,26,73]. Our current findings align with these studies, as we demonstrated that INPP5E affects the recruitment of CD3ζ but not Lck to the immune synapse. Since PI(4,5)P$_2$ is known to be crucial for regulating actin dynamics and vesicle trafficking at the immune synapse formation by binding to actin-regulating proteins, it would be interesting to investigate whether INPP5E influences TCR through vesicular trafficking or actin dynamics at the immune synapse[32].

Modification of phosphoinositides at the plasma membrane also regulates the phosphorylation of proximal TCR components during the formation of the immune synapse[74,75]. CD3ε and CD3ζ contain basic-rich stretches (BRS), allowing these subunits to interact with a series of acidic phospholipids[56,57,76]. Mutations in the BRS sequence result in several defects, including impairing the accumulation of CD3 complex and decreasing the TCR-mediated signaling response after antigen engagement. Moreover, BRS in the CD3ε and CD3ζ have been shown to recruit Lck through ionic interactions to initiate CD3 phosphorylation[39,56,77]. Our studies showed that INPP5E plays an important role in the clustering of CD3ζ, which is essential for T-cell activation. Our finding is consistent with previous research indicating that the interaction between BRS and phosphoinositides is crucial for T-cell activation. INPP5E may prompt CD3ζ phosphorylation via hydrolyzing PI(4,5)P2 at the plasma membrane, providing a preferable phosphoinositide microenvironment for the CD3 subunits to be dissociated from the plasma membrane and interact with Lck, and thus promoting the phosphorylation of ZAP-70 and PLCγ1. Notably, INPP5E may not modulate the Lck activity since pTyr 394 level after the anti-CD3/CD28 engagement was not affected when silencing INPP5E in Jurkat cells. As

previous research has shown that phosphoinositides at the plasma membrane regulate Lck activities by interacting with its SH2 domain[78], further studies are needed to clarify how INPP5E interplays with CD3ζ and Lck.

Changes of phosphoinositide at the immune synapse appear to be a mirror image of those that have been demonstrated in primary cilia[32,33]. INPP5E accumulates in the primary cilia, hydrolyzing PI(4,5)P2 of the cilia and forming an environment for the trafficking and signaling of ciliary proteins[28,79–81]. The immune synapse indeed accommodates some activities similar to those of the primary cilia[13,82,83], while results from us and others suggest that there are still different mechanisms involved in these two structures. For example, TCR recruitment, especially at the early stage, does not require mother centriole polarization and docking. We show that when CD3ζ is recruited to the T-cell-APC contact site during antigen engagement, not all the centrioles in T cells are polarized to the immune synapse. While INPP5E is recruited to the immune synapse, centrioles are not necessarily polarized to the cell contact site in some conjugates, suggesting that the accumulation of INPP5E can be independent of the centriole docking. Next, INPP5E ciliary targeting is mediated by a combination of farnesylation and binding to ARL13B, which is at the C-terminal of INPP5E[41,84]. Although INPP5E interacts with CD3ζ via its C-terminal portion, however, it utilizes the N-terminal proline-rich domain to target the immune synapse. The findings align with the published study, as they provide evidence that INPP5E does not localize to the immune synapse through ciliary targeting signals[85]. Together, ciliary machinery seems to be present at the immune synapse, although detailed activities somewhat differ from each other.

Jurkat T-cell line has been widely used in studying TCR signaling and function. However, the lack of PTEN phosphatase in Jurkat cells can be a strong caveat, especially in studying phosphoinositide metabolism[86]. To strengthen our conclusion, we examined the influence of INPP5E in *Il-2* expression by purifying mouse primary pan T cells. Similar to the results in Jurkat cells, *Il-2* mRNA level was significantly reduced in *Inpp5e*-deficient pan T cells. These results confirmed that INPP5E plays a critical role in T-cell activation both in vitro and ex vivo, and functions of INPP5E in Jurkat are not influenced by other alterations in phosphoinositide signaling, although further studies remain needed to understand the detailed mechanism.

In conclusion, we show that INPP5E regulates CD3ζ accumulation by promoting phosphoinositide dephosphorylation at the immune synapse. The changes of the phosphoinositide may prompt the ITAM motifs of CD3ζ to be phosphorylated by Lck

and activate ZAP-70. The interplay between INPP5E and CD3ζ facilitates cytokine secretion and T-cell activation. Our study unveils how ciliary-enriched phosphatase INPP5E plays an important role in immune synapses and regulates the TCR signaling cascades.

## Methods

**Cells**. Jurkat, Clone E6-1 and Raji cells were purchased from BCRC (#60424 and #60116; Bioresource Collection and Research Center, Hsinchu, Taiwan). Cells were maintained in RPMI 1640 (31800-022, Gibco, Thermo Fisher Scientific, Waltham, MA) medium supplemented with 10% FBS (10437-028, Gibco), 2 mM GlutaMAX (35050061, Gibco), and 1% penicillin/ streptomycin (15070063, Gibco). HEK293T and 293FT cells were cultured in DMEM containing 10% FBS and 1% penicillin/ streptomycin. The identity of Jurkat and Raji cells has been authenticated using STR Profiling Analysis by ATCC with a 100% match. A mycoplasma contamination test was performed by collecting cell culture medium without antibiotics for 24 h and processing this medium by PCR using EZ-PCR mycoplasma detection kit (EZ-PCR Mycoplasma Detection Kit, 20-700-20, Biological Industries, CT, USA), according to the manufacturer's protocol. These cell lines are not in the list of commonly misidentified cell lines maintained by the International Cell Line Authentication Committee. For mouse primary T cells, cells were cultured in RPMI supplemented with 10% heat-inactivated FBS, non-essential amino acids (NEAA, 11140050), sodium pyruvate (11360070), 10 mM HEPES (15630080), 2 mM GlutaMAX, 50uM β- Mercaptoethanol (21985-023), and 1% penicillin–streptomycin (all from Thermo Fisher Scientific).

**Plasmids**. Jurkat cDNA library was generated by SuperScript IV Reverse Transcriptase (Thermo Fisher), and open reading frames (ORFs) for human CD3ζ and Lck were obtained from this library. CD3ζ was subcloned into pEGFP-N1 vector (Takara Bio, Mountain View, CA), and Lck was subcloned into pEF3 vector, which was constructed by replacing CMV promoter on the pcDNA3 vector into EF1α promoter. The ORFs of CD3ζ and Lck were first generated by KOD-Plus-Neo polymerase (TOYOBO, Japan) and then constructed using standard molecular biology techniques. C-terminal Myc-tagged human ZAP-70 plasmid was purchased from Sino Biological (HG10116-CY, China). PH-PLCδ-GFP, PH-AKT-GFP, and GFP-PIPKIγ plasmid were generous gifts from F.-J. Lee (National Taiwan University, Taiwan), and ORF of PIPKIγ was subcloned into the pmCherry-C1 vector. Tubby-GFP plasmid was a kind gift from T. Balla (National Institutes of Health, Bethesda, MD). pSS-FS plasmids, including N-terminal Flag-tagged INPP5E and the series truncation constructs, were kindly provided by S. Seo (University of Iowa, IA).

The targeting sequence of INPP5E is at exon 1 of INPP5E (5′-TGAGATTTGCCTTAGTTTG-3′). For generating mouse shINPP5E plasmid, the targeting sequence of INPP5E was first cloned into the pLKO cloning vector (pLKO_TRC001) via PCR cloning and then subcloned into the pBabe-puro3-EGFP (modified from pBabe-puro3 vector). The pLKO cloning vector was obtained from the RNAi Core of Academia Sinica (http://rnai.genmed.sinica.edu.tw/index).

**Antibodies and reagents**. The indicated antibodies used for immunoblotting (IB) and immunofluorescence (IF) in this study include: FLAG M2 (F1804; 1:500 IF) and γ-tubulin (T6557; 1:500 IF) from Sigma-Aldrich; INPP5E (STJ190490 ; 1:500 IB), pCD3ζ$^{Y83}$ (ab68236; 1:1000 IB, 1:200 IF), and CEP290 (ab84870; 1:200 IF) from Abcam; CD3 (OKT3, 317302, 1:500 IF) from BioLegend; CD3ζ (sc-1239; 1:2000 IB, 1:500 IF), α-tubulin (sc-

32293; 1:10000 IB), GM130 (sc-16268; 1:500 IF), GAPDH (sc-365062; 1:2000 IB) and normal rabbit IgG (sc-2027) from Santa Cruz; NFκB (#8242; 1:2000 IB), pZAP-70$^{Y493}$ (#2704; 1:1000 IB, 1:200 IF), PLCγ1 (#5690,1:1000 IB), pPLCγ1$^{Y783}$ (#2821, 1:1000 IB), Lck (#2984, 1:1000 IB), Lck$^{Y505}$ (#2751, 1:1000 IB), and Src$^{Y416}$ (#6943, 1:2000 IB) from Cell Signaling Technology; ZAP-70 (1 × 17371; 1:500 IB) from Genetex; Myc-Tag (AE009; 1:10000 IB) from ABclonal; Centrin (04-1624; 1:500 IF) from Millipore; mCherry (PA5-34974, 1:1000 IF), GFP (A6455; 1:1000 IF, 1:10000 IB), and DYKDDDDK (PA1-984B; 1:2000 IB) from Invitrogen; Lamin B1 (66095-1-Ig; 1:2000 IB), CEP97 (22050-1-AP; 1:200 IF), CEP164 (22227-1-AP; 1:200 IF), RPGRIP1L (55160-1-AP; 1:200 IF), ARL13B (17711-1-AP; 1:200 IF), and INPP5E (17797-1-AP, 1:200 IF) from Proteintech. CEP164 (1F3G10, 1:2000 IF) was kindly provided by C. Morrison (National University of Ireland, Galway).

Staphylococcus enterotoxin E (SEE) was from Toxin Technologies. CellTracker 7-amino-4-chloromethylcoumarin (CMAC, C2110), ProLong Diamond Antifade Mountant (P36965), Alexa Fluor™ 647 Phalloidin (A22287) and highly cross-adsorbed fluorochrome-conjugated secondary antibodies were from Thermo Fisher Scientific. Horseradish peroxidase (HRP)-conjugated secondary antibodies were from Biolegend (San Diego, CA). Conformation-specific mouse anti-rabbit IgG (#3678) was from Cell Signaling Technology (Danvers, MA). Propidium iodide (P4864), bovine serum albumin (A9647) and poly-L-lysine (P1274) were from Sigma-Aldrich (St. Louis, MO). Protein G Agarose beads were from Agarose Bead Technologies (Doral, FL). Cy3B maleimide was from GE Healthcare (Pittsburgh, PA). Pan T-cell isolation kit II mouse (130-095-130), MACS buffer (130-091-221), and LS column (130-042-401) were from Miltenyi Biotec.

**Mice**. Both male and female C57BL/6J mice, aged 8–10 weeks, were originally purchased from the National Laboratory Animal Center and maintained in the specific-pathogen-free (SPF) facility at the animal center of National Yang Ming Chiao Tung University (NYCU). All experimental procedures of animal studies were approved and performed in accordance with the Institutional Animal Care and Use Committee guidelines of NYCU (IACUC #1090115).

**Purification and activation of murine primary T cells**. For the purification of murine primary T cells, single-cell suspension was prepared from the spleen. Red blood cells were lysed by ACK buffer (150 mM NH$_4$Cl, 10 mM KHCO$_3$, 0.1 mM EDTA) for 2 min at room temperature, followed by PBS wash. The resulting cell suspension was subjected to T-cell isolation using mouse Pan T-cell isolation kit II (Miltenyi Biotec) following the manufacture's protocol. The purity of isolated T cells was confirmed by staining with APC anti-mouse TCRβ (H57-597, 109212, Biolegend) and PE/Cyanine7 anti-mouse CD45 (30-F11, 103113, Biolegend). For T-cell activation, purified T cells were seeded and incubated with DynaBeads Mouse T-Activator CD3/CD28 (11456D, Thermo Fisher Scientific) for total 96 h. Gating strategies are described in Supplementary Fig. 8.

**Isolation of primary human CD4$^+$ T cells**. All other studies with non-diseased primary human cells were performed on samples obtained from 201706119RIND following protocols approved by Institutional Review Board. Blood from healthy donors were obtained after informed consent. Peripheral blood from healthy volunteers was collected using a Vacutainer® blood tube. After removing the plasma, peripheral blood mononuclear cells (PBMC) were isolated using Ficoll-Paque PLUS (GE Healthcare™)

according to the manufacturer's instructions. RBC lysis buffer (containing 155 mM $NH_4Cl$, 10 mM $KHCO_3$, 0.1 mM EDTA) was used to lyse red blood cells if necessary. $CD4^+$ T cells from PBMC were stained with APC anti-CD3 (UCHT1, 300412, Biolegend) and PE anti-CD4 (RPA-T4, 300500, Biolegend) and purified using a BD FACSAria™ cell sorter (BD Biosciences). Gating strategies are described in Supplementary Fig. 8.

**Retrovirus production and infection.** $5 \times 10^5$ of 293FT cells were plated on 6-cm dishes and transfected using T-Pro NTR II transfection reagents (T-Pro Biotechnology, Taiwan) with the following plasmids: 2 µg of VSV-G, 3 µg pCMV-gag-pol and 5 µg of the pBabe-puro3-EGFP based constructs. The supernatant containing viral particles was harvested 24 and 48 h after transfection. Virus-containing media were first centrifuged at 1000 rpm for 5 min to remove the cell debris, and the supernatants were centrifuged for 12 h at $6000 \times g$ at 4 °C to concentrate the virus. Viral pellets were resuspended with 1 mL of fresh T-cell medium. T cells (24 h after activated by Dynabeads) were replaced with viral supernatants containing 8 µg/ml polybrene and then centrifuged for one hour at 2000 rpm at room temperature to enhance transduction. Viral supernatants were replaced with fresh T-cell medium 24 h after transduction (48-h post activation).

**RNA extraction, reverse transcription, and PCR/qPCR.** Transduced primary T cells were sorted by fluorescence-activated cell sorting (FACS) for retroviral reporter expression (viable and GFP positive population) 72 h after transduction. Sorted cells were directly processed using the Single Cell-to-CT™ qRT-PCR Kit (Ambion, 4458236) according to the manufacturer's guidelines. qPCR analysis was performed by using the Luna® Universal One-Step RT-qPCR Kit (NEB, E3005S) according to the manufacturer's instructions. *Rpl19* gene was used as the loading control. All reactions were done in triplicates. Primer sequences were listed: m*Il-2* F: 5-ATCAGCAATATCAGAGTAACTGTTGTA-3′; m*Il-2* R: 5′-CATCTCCTCAGAAAGTCCACC-3′; m*Inpp5e* F: 5′-GATCTTTCAGCCTTCTGGCCC-3′; m*Inpp5e* R: 5′-GAGAGCCATGTTTCGGTCTG-3′; m*Rpl19* F: 5′-GAAATCGCCAATGCCAACTC-3′; m*Rpl19* R: 5′- TCCTTGGTCTTAGACCTGCG-3′.

**Transient transfection.** For siRNA knockdown experiments, two transfections of $10^6$ Jurkat cells were performed at a 24-h our interval using 100 pmol of either control (5′-UUCUCCGAACGUGUCACGU-3′) or *INPP5E* (5′-GGAAUUAAAAGACGGAUUU-3′; J-020852-05, ON-TARGETplus, GE Dharmacon) siRNA with 100-µl tips from Neon Transfection System (Thermo Fisher) (1,400 V, 10 ms, four pulses). For plasmid transfection, 1 µg of DNA was electroporated in $2.5 \times 10^5$ Jurkat cells 48 h after the first siRNA transfection, using 10-µl tips from Neon Transfection System (1325 V, 10 ms, three pulses). Cells were harvested and processed for assay 24 h after DNA transfection or 72 h after the first siRNA transfection[40]. For HEK293T, cells were transfected using PolyJet transfection reagents (SignaGen Laboratories, Rockville, MD). $8 \times 10^5$ cells were plated on a 3-cm plate overnight. Cells were transfected with 1 µg plasmid according to the manufacturer's instructions and harvested 24 h after transfection. The plasmid pCMV-Tag 4A was used as a Flag tag vector control.

**Cell activation.** $1 \times 10^6$ (for immunoblot) or $1.8 \times 10^7$ (for immunoprecipitation) Jurkat Cells were incubated with anti-CD3 (1 µg/ml, OKT3, 317325, BioLegend) and anti-CD28 (1 µg/ml, cd28.2, 302902, BioLegend) on ice for 10 min. Antibodies were then crosslinked with donkey anti-mouse IgG (5 µg/ml, 715-005-

150, Jackson Immunoresearch, West Grove, PA) on ice for another 10 min. Cells were then activated at 37 °C for the indicated times, and the activation was stopped by placing the cells on ice immediately and diluting them with 1 ml cold PBS. Cells were lysed in the lysis buffer supplemented with protease and phosphatase inhibitors, and lysates were centrifuged at $14,000 \times g$ for 10 min. Protein concentration was quantified by the Bradford assay (5000006, Bio-Rad, Hercules, CA), and an equal amount of proteins were boiled in the SDS sample buffer for immunoblotting.

**Coverslip preparation.** 12-mm (0111520) or 18-mm (0111580; Paul Marienfeld, Lauda-Königshofen, Germany) cover glasses were coated with 0.02% poly-L-lysine for one hour at room temperature (RT) and washed twice with distilled water. Cover glasses were dried and used on the same day.

**T-cell staining and T-cell-APC conjugate preparation.** For T-cell-APC conjugates of immune synapse experiments, Raji cells ($10^5$ per sample, used as APCs) were loaded for one hour with 1 µg/ml SEE and labeled with 10 µM of the CMAC dye for the last 20 min in 100-µl serum-free RPMI (SFM) at 37 °C. Cells were washed with warm PBS three times and suspended in SFM. When washing Raji cells, Jurkat cells ($10^5$ per sample) were allowed to adhere to the coated coverslips for 7 min at 37 °C. After that, Raji cells were added to the adhered Jurkat cells for the indicated time. Cells were then fixed in methanol at −20 °C for 10 min (for CEP164, CEP97, and ARL13B) or pre-fixed in 2% PFA in PTEM buffer (50 mM Pipes pH 6.8, 25 mM HEPES, 10 mM EGTA, 10 mM $MgCl_2$, and 0.1% Triton X-100) at RT for 10 min before methanol fixation (for CEP290, RPGRIP1L, and INPP5E). To detect INPP5E and centriole markers in Supplementary Fig. 1, adhered Jurkat cells were pre-extracted with PTEM buffer, either with or without Triton X-100, for 15 s. Cells were then fixed in methanol at −20 °C for 10 min. Fixed samples were imaged with epifluorescence microscopy (Olympus X81) with a 100× (NA = 1.30) oil-immersion objective.

For phosphoinoside detection, $2.5 \times 10^5$ siCtrl, si*INPP5E*, or wild-type Jurkat cells were electroporated with indicated plasmids. After 24 h, Jurkat cells ($1 \times 10^5$ per 20-µl SFM) were first plated for 7 min at 37 °C on 12-mm precoated coverslips. $1 \times 10^5$ SEE-loaded Raji cells (in 20-µl SFM) were then added to each coverslip and co-cultured for 10 min. Cells were immediately fixed for 10 min at RT by adding 40 µl 8% PFA, permeabilized with PBST, and blocked as described above. For INPP5E domain mapping, $2.5 \times 10^5$ Jurkat cells were electroporated with series truncations 24 h before experiments. T-cell-APC conjugates were prepared as mentioned in phosphoinoside detection but with final co-culturing for 20 min. Cells were immediately fixed for 10 min at RT by adding an additional 4% PFA (final 2%), permeabilized with methanol for 20 min, and blocked as described above. For CD3ζ-GFP reconstitution and phosphorylated proteins, 40-µl conjugates (co-cultured for 10 min) were fixed for 10 min at RT by adding 40-µl 8% PFA, permeabilized with PBST, and blocked as mentioned above. Confocal images were carried out on a Zeiss LSM780 using a 100× objective (NA = 1.40). 1 Airy Unit was set as a pinhole for each channel. For Z-series images, sections were set at 0.3-µm intervals.

**T-cell spreading preparation.** For spreading T cells on a planar surface, 18-mm precoated coverslips were coated for 2 h at 37 °C with 1-µg/ml anti-CD3/CD28. Cover glasses were washed three times with warm PBS and kept for 10 min in 37 °C. $10^5$ Jurkat cells were then incubated on coverslips for 10 min before being fixed with 4% PFA[40]. Single plane images were acquired on the

confocal microscope (for F-actin spreading) or total internal reflection fluorescence (TIRF) microscope (for INPP5E and CD3ζ spreading; ELYRA, Zeiss) with a 100× (NA = 1.46) oil-immersion objective lens.

**Immunostaining**. After fixation, cells were washed for 5 min with PBS, permeabilized for 10 min with 0.1% PBST (PBS with 0.1% Triton X-100), and blocked with 3% BSA/PBST for 30 min. Primary antibodies at indicated dilution were prepared with blocking buffer and incubated with cells for one hour. Following three washes in PBST, the cells underwent an additional 1-h incubation with Alexa Fluor-labeled secondary antibodies (diluted at 1:500) and phalloidin dye (diluted at 1:200). Finally, cells were washed three times with PBST and mounted with ProLong Diamond Antifade Mountant.

**Image analysis**. ImageJ, Imaris8.2 (Oxford Instruments), and ZEN (Zeiss) software were utilized for image processing and analysis. For control and si*INPP5E* cell comparison, the same acquisition settings and thresholds were used for each image.

For assays related to the recruitment/enrichment of proteins to the synapses at the T-cell-APC conjugates, Jurkat cells with similar surface CD3ζ-GFP and Flag-INPP5E levels were chosen for comparing the recruitment ability. For phosphorylated protein enrichment, T-cell-APC conjugates from each experiment were randomly selected for processing. Recruitment/enrichment index was measured by a method developed by previous literature and graphed in Supplementary Fig. 2B[39,64,87,88]. Specifically, equal area was selected at the cell-cell contact site (proximal), a region of the T-cell opposite from the cell contact site (distal), and a background area outside of the cell (BG). The recruitment index was calculated with the formula: [MFI at the proximal site – BG]/ [MFI at the distal site − BG]. Quantification of MFI was performed with Image J. For recruitment of INPP5E truncation, we scored each conjugate as positive when most of the Flag signals were accumulated at the cell contact.

For recruitment of endogenous INPP5E to mouse pan T cells. The anti-CD3/CD28-coated beads were used to activate immune synapse formation. The INPP5E signals that were detected at T cells-beads conjugation sites were counted as positive cells.

For recruitment of endogenous INPP5E to the pseudo-synapse on TIRF images, spreading cells were chosen randomly from each experiment, and the area for measurement was selected by the contour of CD3ζ. Image J was used to quantify the MFI.

For the quantification of phosphoinositide and F-actin clearance, Z-stack confocal images were reconstructed into 3D images by Imaris software, and the en face view at the contact site was used for analysis. Clearance ability was assessed following previous studies[89,90]. Briefly, 'cleared' was scored when a continuous GFP ring or F-actin was formed at the center of the synapses; any interruption at the center was classified 'not cleared'.

**Live-cell microscopy**. $2.5 \times 10^5$ siCtrl and si*INPP5E* Jurkat cells (48 h post first siRNA electroporation) were electroporated with 0.5 μg of LifeAct-TagRFP and 0.5 μg of indicated lipid probes using Neon 10-μl tips and R buffer. After 24 h, $1 \times 10^5$ Jurkat cells (in 20 μl SFM) were first plated for 7 min at 37 °C per well on 8-well chambered borosilicate coverglass (Lab-Tek). $1 \times 10^5$ SEE-CMAC-loaded Raji cells (in 20 μl SFM) were then added dropwise to T cells, and imaging were started in 3 min. Interaction of T-cell-APCs were monitored in the chamber maintained at 37°C (Carl Zeiss) on an using Cell Observer SD confocal microscopy equipped with YOKOGAWA CSU-X1 system (Carl Zeiss). Z stacks with 0.8 μm intervals were collected every 30 s with

fluorophores excited at 405, 488, and 561 nm in each z-plane. Images were analyzed with ZEN.

**3D-SIM**. Three-dimensional structured illumination microscopy (3D-SIM) was preformed following the standard protocol. In brief, Jurkat cells were conjugated with SEE-loaded Raji cells at indicated time points. Cells were fixed and stained with anti-CD45 (Biotium Cat#0313) and INPP5E (Proteintech). After washing, the cells were incubated with Alexa Fluor 488- and Alexa Fluor 555-conjugated secondary antibodies (Invitrogen). The 3D-SIM images were preformed using a Zeiss ELYRA PS.1 LSM780 system equipped with a Plan Apochromat 63×/1.4NA oil-immersion objective. Z stacks with an interval of 110 nm were used to scan the whole cells. The raw images were reconstructed using ZEN software under the default parameters.

**Immunoblotting**. Cells were washed with PBS twice and lysed in the ice-cold lysis buffer (20 mM Tris-HCl, pH 7.5, 150 mM NaCl, 1 mM EDTA, 1% NP-40) that contains protease inhibitors (Sigma-Aldrich, 11697498001) and PhosSTOP inhibitors (Sigma-Aldrich, 4906845001). After centrifuging at $14,000 \times g$ for 10 min at 4 °C to remove cell debris, supernatants was taken and protein concentrations were determined by the Bradford protein assay (Bio-Rad). Equal amounts of proteins were mixed with the SDS sample buffer, boiled at 95 °C for 5 min, and resolved by SDS-PAGE. The separated proteins were then transferred to the PVDF membranes (0.45 μm Immobilon®-E; Merck Millipore). Membranes were blocked either with 2.5% nonfat milk in TBST (50 mM Tris, pH 7.5, 150 mM NaCl, and 0.05% Tween-20) or 3% BSA (for phospho antibodies) for 1 h at RT and incubated with primary antibodies overnight at 4 °C. Blots were washed three times with TBST and incubated with HRP-conjugated secondary antibodies (Biolegend) for 1 h at RT. After washing three times with TBST, proteins were visualized with Western Lightning Plus substrate (PerkinElmer, NEL103E001EA). Uncropped western blot images are provided in Supplementary Figs. 6 and 7.

**Cell fractionation**. Jurkat cells were first activated as described and immediately washed with ice-cold PBS twice. Cells were then fractionated with NE-PER™ Nuclear and Cytoplasmic Extraction Reagents (78833, Thermo Fisher Scientific) according to the instruction from the supplier. Cytosol and nucleus fractions were mixed with sample buffer and analyzed by western blot. GAPDH (cytosol marker) and Lamin B (nucleus marker) were used to verify the purity of fractionation.

**Immunoprecipitation**. Jurkat cells were first activated as described in the "Cell Activation" method section. After activation, Jurkat cells were immediately mixed with ice-cold 5× lysis buffer (100 mM Tris-HCl, pH 7.5, 750 mM NaCl, 5 mM EDTA, 5% NP-40) supplied with protease inhibitors. Cells were then centrifuged at $14,000 \times g$ for 10 min at 4 °C to remove cell debris. Cell lysates were first treated with 20-μl protein G agarose beads for one hour to remove antibodies for activation. Eight hundred micrograms lysates were suspended in 400-μl lysis buffer and immunoprecipitation was performed by using 1 μg anti-INPP5E antibody (Proteintech, Rosemont, IL) at 4 °C overnight, and incubated with 20 μl protein G agarose beads for additional one hour on the next day. The beads were washed three times with the lysis buffer, eluted by boiling in sample buffer and resolved by SDS-PAGE.

For immunoprecipitation in HEK293T cells, $8 \times 10^5$ cells were transfected as described above. After 24-h transfection, cells were lysed in the ice-cold lysis buffer (20 mM Tris-HCl, pH 7.5, 150 mM NaCl, 1 mM EDTA, 1% NP-40) supplied with protease

inhibitors. Five hundred micrograms of pre-cleared lysates were first incubated with 1-µg anti-FLAG antibody (Sigma-Aldrich) overnight followed by incubating with 20-µl protein G agarose beads for another hour. The beads were washed three times with the lysis buffer with 0.01% glycerol, eluted by sample buffer, and resolved by SDS-PAGE.

**Flow cytometry**. For analysis of surface expression of CD25 and CD40L, Jurkat cells were stimulated with anti-CD3/CD28 (1 µg/ml) antibodies for indicated time points. Cells were collected, washed with PBS and stained with Zombie Violet™ Fixable Viability (423113, Biolegend) for 10 min on ice according to the manual, and sequentially stained with PE-CF594 anti-CD25 (562403, BD bioscience) and AF488 anti-CD40L (310815, Biolegend) in FACS buffer (2% FBS, 2 mM EDTA in 1× PBS). Cells were stained for 20 min on ice, washed twice with FACS buffer and analyzed by flow cytometry. The gating strategies have been outlined in Supplementary Fig. 8.

**ELISA**. For measuring the IL-2 level in T cells that were activated by anti-CD3/CD28 antibodies, $1 \times 10^5$ Jurkat cells were seeded in 96-well plates and treated with anti-CD3/CD28 (1 µg/ml) antibodies for 24 h. For measuring the IL-2 level in T-cell-APC conjugates, $1 \times 10^5$ Jurkat cells were conjugated with the same amount of SEE-loaded Raji cells (SEE concentration: 1 µg/ml) in 96-well plates for 24 h, and cells were collected. Cells were centrifuged at $2000 \times g$ for 10 min at RT, and the supernatants were collected. IL-2 was quantitated with ELISA kits (OptEIA, 555190, BD bioscience). The assays were carried out in accordance with the manufacturer's instructions.

**Statistics and reproducibility**. Statistical analyses were conducted using GraphPad Prism 8 software. Experimental design, utilized statistical methods, and significance levels in various data analysis within this study can be found in the respective figure legends and methods. The presented data are the results from at least three independent experiments. The statistical difference was mostly analyzed by unpaired t-tests for two-group comparisons except Fig. 6h with paired t-tests. In cases involving more than two groups, one-way ANOVA followed by Dunnett multiple comparison test was utilized. Differences with two-tailed p-values < 0.05 were considered statistically significant. The error bars represent the mean along with standard deviation (SD) for bar graph.

**Reporting summary**. Further information on research design is available in the Nature Portfolio Reporting Summary linked to this article.

## Data availability

The source data behind the graphs in the main and supplementary figures are available in Supplementary Data 1 and 2. Uncropped western blot images are provided in Supplementary Figs. 6 and 7. Gating strategies are described in Supplementary Fig. 8. All other data are available from the corresponding author upon request.

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

## Acknowledgements

We thank Dr. Jen Liou (The University of Texas Southwestern Medical Center, Dallas, TX) for the discussion and critical comments. We thank Sue-Ping Lee (Imaging Core at Institute of Molecular Biology, Academia Sinica) for assisting with the technical assistance of TIRF and 3D-SIM microscopy. We also thank the staff of Technology Commons

(College of Life Science, National Taiwan University) for help with confocal microscopy and flow cytometry. This work was supported by the National Science and Technology Council, Taiwan (NSTC 107-2313-B-001-009, 108-2313-B-001-003) and National Taiwan University and Academia Sinica Innovative Joint Program Grant (NTU-SINICA-108L104303) to J.C.L.; by the National Health Research Institutes (NHRI-EX109-10610BC) and National Taiwan University and Academia Sinica Innovative Joint Program (109L104303) to H.C.T.; by the NSTC 109-2628-B-010-016 and Cancer Progression Research Center NYCU, from the Higher Education Sprout Project by MOE to C.L.H.; by the NSTC 110-2326-B-A49A-503-MY3, 111-2628-B-A49A-016, and 112-2628-B-A49-009-MY3 to W.J.W.

## Author contributions

T.Y.C., H.C.T., Y.W.L., C.L.H., W.J.W., and J.C.L. designed research; T.Y.C., C.H.L., Y.H.L., Y.D.L., S.S.L., Y.T.F., S.Y.H., W.S.H., P.Y.T., F.H.Y., and Y.C.W. performed the experiments; T.Y.C., C.H.L., Y.H.L., Y.D.L., S.S.L., Y.T.F., S.Y.H., W.S.H., P.Y.T. and W.M.C. analyzed data; and T.Y.C., H.C.T., C.L.H., W.J.W., and J.C.L. wrote the paper.

## Competing interests

J.C.L. is the CEO and Board Director and owns stock of Syncell Inc. W.M.C. is an employee and owns stock of Syncell Inc. All other authors declare no competing interests.
