## [Peer Review File · Communications Biology]

Reviewers' comments:

Reviewer #1 (Remarks to the Author):

In this manuscript, Chiu et al present interesting findings examining the role of inositol polyphosphate-5-phosphatase E (INPP5E) in T cell activation. The figures and results are generally well presented. However, for these results to be appreciated, the introduction should better explain current concepts of the functions and regulation of phosphoinositides and the cytoskeleton/microtubules in the immunological synapse during T cell activation. For example, although Gawden-Bone et al Immunity 2018 and Frontiers in Immunology 2019 are cited (references 18 and 58), the manuscript does not explain the findings and hypotheses presented in these papers, which are highly relevant to the current manuscript. The discussion will need to be revised to align with the revised introduction. Overall, the findings are interesting, but substantial revisions are required to frame the results in a way that they will influence thinking the field.

The main claim of the paper is that "INPP5E is a new player in phosphoinositide manipulation at the synapse, controlling the TCR signaling cascade." This was predicted in considerable detail, but not proved, by Gawden-Bone and Griffiths in Frontiers in Immunology 2019. So, the findings appear to be novel and should be of interest to the T cell community if they are suitably framed.

Below is a list of notes to guide some of the more minor revisions required.

Abstract

Line 37 As "antigen-specific" implies TCR interactions with peptide-MHC ligands, here it should be replaced by "superantigen-mediated", or similar. The same comment applies to line 103 in Results.

Introduction

Line 49 I recommend deleting "allowing the conduction of proximal TCR signaling". Although it is clear that a cSMAC forms, it is still debated whether the cSMAC is the site of signaling-competent TCRs or the site of TCR removal from the immunological synapse (see papers by Michael Dustin).

Line 54 Each of the four subunits of CD3 has at least one ITAM, so it is unclear why only two subunits are mentioned.

Line 65 Quotation marks should be used for "a frustrated cilium".

Results

Line 158 As the CTS (ciliary targeting sequence) is mentioned in the text, the CTS probably should be annotated in Figure 3a.

Line 179 The text indicates certain proteins were individually co-expressed with INPP5E in HEK293T cells. Yet, in Figure 3b, the labels on the right gel do not align with this statement in the Results. Are the labels wrong? In addition, would it be clearer to change the right gel in Figure 3b to Figure 3c?

Line 200 "Interestingly," could be replaced with "In the presence of SEE,"

Line 215 The description of "PH-PLCdelta1" is somewhat unclear. Shouldn't it include "the pleckstrin homology (PH) domain of PLCdelta1 fused to GFP"?

Line 221 "the distribution of PH-PLCdelta were" should be "the distribution of PH-PLCdelta was"

Line 226 "a clear clearance" sounds a bit unusual

Line 241 I do not understand why the text says, "under PIP2 ablation". If I understand Supp fig 4a,b, which use Tubby-EGFP as a PIP2 sensor, PIPKIgamma expression does not ablate PIP2, but it appears to prevent the clearance of PIP2 from the SMAC.

Line 245 I find "negative regulation of PI(4,5)P2" ambiguous. To me, "abnormal PI(4,5)P2 levels and/or distribution" would make sense.

Line 250 "stained with known" should be "stained with antibodies specific for known"

Line 258 The sentence ending on line 258 is confusing. Is "(Supplementary Fig. 4c)" the correct figure panel? Should the citation to the figure panel be positioned earlier in the sentence?

Line 259 "these data suggests" should be "these data suggest"

Line 272 For clarity, I recommend deleting "the phosphorylation of CD3zeta and ZAP-70 did not appear to be affected by Lck, as"

Discussion

Lines 296-299 are difficult to understand, even after consulting reference 37.

Materials and Methods

Line 357 "The identity of the cell line..." Which cell line(s) does this statement refer to?

Lines 370 - 374 Please clarify what the described reagents are. Are they plasmids?

Line 401 What data relates to the "Pan T cell isolation kit II mouse"?

Line 484 "Person's" should be "Pearson's"

Line 505 "accessed" should be "assessed"

Figure Legends

Line 746 Figure 1 legend mentions arrows, but there are no arrows on the images.

Line 764 and others The legend does not need to say "ns, not significant" when a statistically significant difference is shown.

Line 770 In the 5 min and 10 min images, the bright red signal (INPP5E) is mainly in the CMAC+ cell. Previously, CMAC was used to stain the APC. If the experimental approach in Figure 2h is different, then this should be explained in the Figure Legend.

Line 783 The word "green" seems to be an error here

Line 827 Figure 5c legend mentions arrows, but there are no arrows on the images.

Lines 856 - 860 It may be informative to test whether INPP5E-specific siRNA affects IL-2 secretion during "weak" stimulation, "strong" stimulation, or both. Were different concentrations of anti-CD3/CD28 and/or SEE tested? The Figure Legend also does not state how many technical replicates were performed in each experiment.

Supp Fig 1a In the lower panels, what do the arrows point to?

Supp Fig 1c Do the upper and lower panels show cells analysed in the absence and presence of SEE, respectively? If so, it should be explained in the figure legend and annotated on the figure.

Supp Fig 3b Should the lower panels be labelled "Input"?

Reviewer #2 (Remarks to the Author):

In this study Chiu and colleagues identify for the first time a role for the ciliary protein INPP5E in the formation of the immune synapse. Using several experimental approaches, the authors show that this ciliary protein localizes to the IS upon superantigen stimulation. Elegant studies showed that downregulation of INPP5E by siRNA impairs CD3Zeta and Zap70 localization to the immune synapse and then clearance of PIP from the IS is compromised leading to less activation of CD3Zeta but not to Lck which localization to the IS is not compromised. The authors performed extended experimentation using complementary techniques to show the importance of INPP5E in the immune synapse, which at least in their experimental systems seem of clear importance in the T cell APC interaction. The findings are novel and relevant in the immunology field. Yet the models used to test these are a bit limited. First, no experiments in primary T cells have been performed to ensure that this is not an artefact resulted from using a transformed cell line, second, most of the evidence comes from very strong CR interactions driven by CD3 antibodies and superantigens that may not represent most of the antigen specific interactions. Yet the evidence provided regarding INPP5E participation in the IS. The presentation of the data is clear, the text is well written and the discussion very thorough. The statistical analyses are correct and all the data regarding experimental repetitions and number of cells analysed is provided.

Major comments

- All the study is performed on the Jurkatt Cell line which is a very good model but still a transformed cell line. The authors should compare the expression levels of INPP5E in normal T cells from several donors to the Jurkatt cell line and at least show that in primary T cells undergoing an immune synapse INPP5E is also directed to the IS.

-There is little evidence in the overall effect on T cell activation. Please provide data regarding IL-2 production as IL-2 actual concentration and do not normalize the data. Please provide some evidence on early T cell signaling events (NFAT translocation, NFkB and/or AP-1 signaling) and other T cell activation markers (could be CD137, CD69 , CD25,..).

Alvaro Teijeira, 20.09.22, Immunology and immunotherapy department, CIMA, universidad de Navarra. Spain.

Reviewer #3 (Remarks to the Author):

In this manuscript Chiu et al investigate the distribution of cilia-associated proteins in resting and superantigen (SEE)-activated T cells, finding that all the proteins investigated showed a redistribution toward the immunological synapse (IS). The rest of the manuscript focusses on INPP5E, which they show accumulates at the IS of SEE-activated and anti-CD3epsilon/anti-CD28-activated T cells. Using truncation mutants the authors suggest that N-terminal proline-rich domain is responsible for

accumulation of INPP5E at the IS. Using immunoprecipitation the authors also suggest there is a direct interaction between INPP5E and CD3zeta, Lck and ZAP70. In contrast to IS accumulation, the C-terminal domain appears to be more important for the proposed interaction with CD3zeta. Knockdown of INPP5E resulted in reduced accumulation of CD3zeta at the IS and defective clearance of PI(4,5)P2 from the center of the synapse, despite normal clearance of actin from the center of the synapse. INPP5E knockdown also reduced the level of phosphorylated CD3zeta, and ZAP70 in response to TCR and CD28 antibody cross-linking, but did not affect phosphorylated PLCgamma1 or Lck (on the activating Y394 residue), suggesting an effect on proximal signalling efficiency.

Overall, I found the work to be interesting, with many well performed experiments and intriguing observations. My one regret after reading this is that despite the breadth of the results the mechanism underpinning and linking the observations is difficult to glean, although the effect on CD3zeta (and by extension TCR) accumulation and clearance of PI(4,5)P2 from the IS were particularly striking. I would also advise some caution about extending conclusions made about super-antigen stimulated T cells to T cells stimulated conventionally through TCR-agonist peptide MHC interactions. I would also strongly advise caution about extending conclusions about phosphoinositide biology in Jurkats as this cell line has a documented deficiency in PTEN resulting in constitutive membrane localisation and activity of ITK (doi: 10.1128/MCB.20.18.6945-6957.2000.). Some discussion of how this may impact the translation of findings from this study to primary cells should be made in the Discussion section.

I have the following specific comments:

- Fig 2D: CD3zeta and INPP5E do not look colocalised, despite results from pull-downs suggesting they should directly interact. There is also no obvious difference in the accumulation of CD3zeta in the synapse of cells with INPP5E knocked down. Please comment on these discrepancies with other results.
- Despite good INPP5E knockdown shown in the western blot (Fig 2a) the staining intensity appears similar, with the exception that staining in the IS is abrogated specifically (Fig 2b and Fig 4c). This seems to indicate that there is some non-specific binding of the antibody in the cells, which doesn't affect the conclusions but could at least be commented on briefly when describing Fig 2.
- Fig 2F: The CD3epsilon staining looks strangely cytoplasmic in -SEE, this doesn't look right at all, it should look membrane bound with some in the ER and Golgi as they traffick to the surface.
- Fig 3D: There is a lot of information missing in the Fig 3 legend. What does each dot represent in the top panel of Fig 3D? Does each dot represent a single cell? What about the bottom panel? The average of values from the two experiments? If so, how did you do statistics on two values? What statistical analysis was used here and what do the symbols indicate?
- Fig 4G: The example en face inset is very difficult to see, this should be made larger. How many cells are the example live cell images indicative of? Was it only performed once?
- lines 306-307: I find it difficult to understand what is meant by "actively and passively" here. I would suggest "...directly via enzymatic activity, or indirectly via an alternative mechanism" may more accurately convey the meaning.

In particular, I found that many of the conclusions are framed in very strong terms, but the evidence does not fully support some of the statements made. For example:

- Fig 3C: Accumulation at the IS looks very similar in the examples given. From the quantitation it seems there is still significant accumulation without the proline-rich domain, despite it being stated that this domain is "responsible" as is stated for example in the title of the Figure.
- The final section title states that "INPP5E is essential for proximal TCR signalling" (also lines 283-284) but the data show a ~40% reduction in pCD3zeta, a non-significant reduction of pPLCg and a ~30% reduction in IL-2 with knockdown of INPP5E. This indicates that INPP5E may play a role in efficient proximal signalling, but is a long way from demonstrating it is essential.
- lines 296-299: This is not accurate, it is not that TCR diffusion "requires" PI(4,5)P2

dephosphorylation but ref 37 demonstrates that this releases CD3epsilon chains from the membrane resulting in very modest increases in TCR diffusion rates (Fig 3b in ref 37).

- lines 315-316: Although it is stated that "Our studies on INPP5E strongly support the importance of BRS-phosphoinositides interactions in response to T cell activation." there is no direct link shown here between INPP5E regulated PI(4,5)P2 levels and the association of TCR BRS regions with the membrane. This should be restating this as "...results are consistent with...".

We would like to thank the reviewers and editors for the positive notes and critical suggestions. We have revised the manuscript according to the comments. Significant changes are itemized below, followed by our point-by-point responses to address reviewers' comments.

Significant changes in the revised manuscript:

(1) We have revised our introduction and discussion to explain the current concepts regarding the role of phosphoinositides and the cytoskeleton/microtubules at the immune synapse during T cell activation (line 81-92; line 329-347).

(2) We compared the INPP5E levels in T cells from multiple healthy donors to the Jurkat cells. Our results showed that the INPP5E expression levels in primary T cells are comparable to the Jurkat cells (Supplementary Fig. 2c). It supports the use of Jurkat cell line in this study.

(3) We purified mouse pan T cells and induced immune synapse formation by using anti-CD3/CD28 coated beads. Our results showed that INPP5E signals were concentrated at T cells-beads conjugation sites after 20 minutes of conjugation (Supplementary Fig. 2d, e). It indicates that INPP5E is directed to the immune synapse in primary T cells upon T cell activation.

(4) Although challenging, we confirmed the importance of INPP5E in regulating T cell activation using isolated mouse primary pan T cells. We found that *Il-2* mRNA level in *shInpp5e*-transduced pan T cells were significantly reduced in compared with that in the control T cells, confirming the role of INPP5E in T cell activation. The results have been added to Fig. 6h, and the descriptions have been added to line 314-319. Discussion was added to line 382-388. Materials and Methods section is updated by adding contents in line 463-503.

(5) To have a better understanding regarding the overall effects of INPP5E on T cell activation, we first analyzed the influence of INPP5E in early T cell signaling events. We analyzed NF- κ B by monitoring its nuclear translocation. The control and *siINPP5E* Jurkat cells were activated with anti-CD3/CD28 to stimulate the TCR response. The *siINPP5E* cells reduced NF- κ B nuclear translocation compared to control cells (Fig. 6d). We also checked the expression of CD40L since CD40L is expressed on the surface of activated T cells. In compared to the control cells, the percentage of CD40L-expressing cells was significantly reduced in *siINPP5E* Jurkat cells, further demonstrating the importance of INPP5E in coordinating immune responses (Fig. 6e). To determine the impact of INPP5E depletion on T cell effector functions, we measured the surface expression of CD25 and the secretion amount of IL-2 in control and *siINPP5E* Jurkat cells after T cell activation. While INPP5E knockdown only slightly reduced CD25 expression (Supplementary Fig. 5b), IL-2 secretion was significantly attenuated in *siINPP5E* cells compared to control cells (Fig. 6f, g), regardless of whether the cells were activated with superantigen-loaded Raji cells or anti-CD3/CD28, indicating that INPP5E plays an important role in regulating T cell effector functions. Materials and Methods section is updated by adding contents in line 649-654, 675-682.

(6) We have also changed our wording to avoid making any conclusion in very strong statement without enough evidence in the paper.

(7) Thanks to the assistance of people in the lab, the paper revision can be completed smoothly. We added them as the co-authors (Yi-Hsuan Lin, Yun-Di Lai, and Pei-Yuan Tsai) in the revised manuscript.

Reviewer #1 (Remarks to the Author):

In this manuscript, Chiu et al present interesting findings examining the role of inositol polyphosphate-5-phosphatase E (INPP5E) in T cell activation. The figures and results are generally well presented. However, for these results to be appreciated, the introduction should better explain current concepts of the functions and regulation of phosphoinositides and the cytoskeleton/microtubules in the immunological synapse during T cell activation. For example, although Gawden-Bone et al Immunity 2018 and Frontiers in Immunology 2019 are cited (references 18 and 58), the manuscript does not explain the findings and hypotheses presented in these papers, which are highly relevant to the current manuscript. The discussion will need to be revised to align with the revised introduction. Overall, the findings are interesting, but substantial revisions are required to frame the results in a way that they will influence thinking the field. The main claim of the paper is that "INPP5E is a new player in phosphoinositide manipulation at the synapse, controlling the TCR signaling cascade." This was predicted in considerable detail, but not proved, by Gawden-Bone and Griffiths in Frontiers in Immunology 2019. So, the findings appear to be novel and should be of interest to the T cell community if they are suitably framed.

We appreciate the reviewer for supporting the importance of our studies and thanks for nice suggestion. We have included the current concept regarding the functions of phosphoinositides and the cytoskeleton/microtubules in the immunological synapse during T cell activation in the introduction and discussion of revised manuscript.

Introduction (line 81-92)

The regulation of phosphoinositides and actin dynamics is important for immune synapse formation and T cell activation. Phosphoinositides, particularly PI(4,5)P₂, play crucial roles for the formation and maintenance of the immune synapse. Upon T cell receptor engagement with antigen-presenting cells, the phospholipase C-gamma (PLCγ) is rapidly recruited to the immune synapse, which catalyzes the hydrolysis of PI(4,5)P₂ into inositol 1,4,5-trisphosphate (IP3) and diacylglycerol (DAG). PLCγ causes the decrease of PI(4,5)P₂ levels at the immune synapse, which is necessary for the efficient centrosome docking at the synapse and subsequent granule secretion. The dynamic regulation of actin polymerization at the immune synapse facilitates the delivery of activating signals to the T cell that promotes the communication between the T cell and the antigen-presenting cell. Since PI(4,5)P₂ is reported to bind to a variety of actin-regulating proteins, PI(4,5)P₂ coordinates the assembly and disassembly of actin filaments by binding to these actin-regulating proteins.

Discussion (line 329-347)

The TCR/CD3 complex is a crucial receptor complex that is involved in T cell activation and phosphoinositides are important in regulating the dynamics of this complex. Studies have shown that dephosphorylation of PI(4,5)P₂ through overexpressing the plasma membrane-localized inositol polyphosphate-5-phosphatase (Inp54p) in the mouse T-cell hybridoma causes the CD3ε cytoplasmic domain unbinding from the plasma membrane, resulting in very modest increase in TCR diffusion rate. In contrast, primary T cells that overexpress PIP5K, an enzyme involved in PI(4,5)P₂ synthesis, increases the amount of PI(4,5)P₂ at the immune synapse, leading to T cell rigidity. It thus delays the recruitment of TCR complex and impairment of proximal TCR signaling. The conclusion of these studies is consistent with the results in our works. We observe that PI(4,5)P₂ level at the center of the synapse was higher in

siINPP5E Jurkat cells, and the reconstitute CD3 ζ -GFP was less recruited to the immune synapse in these cells, suggesting that the diffusion ability of CD3 ζ at the plasma membrane was decreased. One possibility is that INPP5E directly regulates PI(4,5)P₂ levels via its enzymatic activity. Alternatively, INPP5E could regulate PI(4,5)P₂ indirectly by interacting with other signaling molecules that affect PI(4,5)P₂ levels. It remains to be examined whether INPP5E regulates PI(4,5)P₂ directly via enzymatic activity, or indirectly via an alternative mechanism. Since PI(4,5)P₂ is known to play crucial roles in the regulation of actin dynamics at the immune synapse formation through binding to actin-regulating proteins. Whether INPP5E directly or indirectly affects actin dynamic at synapse represents an interesting topic for future investigation.

Below is a list of notes to guide some of the more minor revisions required.

Abstract

Line 37 As "antigen-specific" implies TCR interactions with peptide-MHC ligands, here it should be replaced by "superantigen-mediated", or similar. The same comment applies to line 103 in Results.

Thanks for nice suggestion. We have changed it to "superantigen-mediated conjugation" in the abstract (line 37) and line 103 (currently in line 115) in results.

Introduction

Line 49 I recommend deleting "allowing the conduction of proximal TCR signaling". Although it is clear that a cSMAC forms, it is still debated whether the cSMAC is the site of signaling-competent TCRs or the site of TCR removal from the immunological synapse (see papers by Michael Dustin).

We agree with the comment and have deleted it from the introduction.

Line 54 Each of the four subunits of CD3 has at least one ITAM, so it is unclear why only two subunits are mentioned.

Thanks for the comment. Indeed, each of the subunits of the CD3 complex (ϵ , γ , δ , and ζ) contains at least one ITAM. The intracellular domains of each of the CD3 chains contain ITAMs that serve as the nucleating point for the intracellular signal transduction machinery upon TCR engagement. The CD3 δ , γ , and ϵ chains each contain one ITAM, and CD3 ζ contains three ITAMs.

According to the published papers, the ϵ and ζ chains of the CD3 complex are reported to be required for proper surface expression and assembly of the TCR complex. Defects in the ϵ and ζ chains can lead to impaired TCR signaling and immunodeficiency.

Although the ϵ and ζ chains of CD3 are reported to be important for TCR signaling, CD3 δ and γ are also important in proper TCR signaling. We thus decided to change the sentence to:

The proximal TCR signaling requires Src family kinase Lck to phosphorylate the immunoreceptor tyrosine-based activation motifs (ITAMs) of CD3. These motifs are docking sites for the tyrosine kinase zeta-chain-associated protein kinase 70 (ZAP-70)" (line53-54).

Ref:

J Immunol. 2009 Jul 15;183(2):1055-64.

J Immunol. 2011 Jun 15;186(12):6839-47.

N Engl J Med . 2006 May 4;354(18):1913-21.

Line 65 Quotation marks should be used for "a frustrated cilium".

Added. Thanks for the suggestion (now in line 64).

Results

Line 158 As the CTS (ciliary targeting sequence) is mentioned in the text, the CTS probably should be annotated in Figure 3a.

We have added the CTS (ciliary targeting sequence) in Fig. 3a.

Line 179 The text indicates certain proteins were individually co-expressed with INPP5E in HEK293T cells. Yet, in Figure 3b, the labels on the right gel do not align with this statement in the Results. Are the labels wrong? In addition, would it be clearer to change the right gel in Figure 3b to Figure 3c?

We really apologize for the confused statement in line 179 and the wrong label in the right gel of figure 4b (current Fig. 4c).

In fig.4c, we aimed to investigate the interaction between INPP5E and three different proteins: CD3 ζ , ZAP70, and Lck. To do this, we co-expressed INPP5E and each of these proteins separately in 293T cells. We then performed an immunoprecipitation of INPP5E from the cell lysates and found that INPP5E could pull down CD3, ZAP70, or Lck individually.

We changed our description in the text to make the statement more clearly. "By co-expressing INPP5E and either CD3 ζ , ZAP70, or Lck separately in 293T cells, INPP5E interacted with these proteins individually, indicating INPP5E directly interacts with CD3 ζ , Lck, and ZAP-70 proteins (Fig. 4c)." (now in line 195-198)

We also corrected our label in figure 4c.

Line 200 "Interestingly," could be replaced with "In the presence of SEE,"

We agree with your suggestion and have changed it to in the presence of SEE (now in line 221).

Line 215 The description of "PH-PLCdelta1" is somewhat unclear. Shouldn't it include "the pleckstrin homology (PH) domain of PLCdelta1 fused to GFP"?

We agree with your suggestion and have correlated it (in line 235-236). Thanks.

Line 221 "the distribution of PH-PLCdelta were" should be "the distribution of PH-PLCdelta was"

Fixed it (now in line 241). Thanks.

Line 226 "a clear clearance" sounds a bit unusual

We agree with your comment and have changed it to " a complete clearance" (now in line 245).

Line 241 I do not understand why the text says, "under PIP2 ablation". If I understand Supp fig 4a,b, which use Tubby-EGFP as a PIP2 sensor, PIPK γ expression does not ablate PIP2, but it appears to prevent the clearance of PIP2 from the SMAC.

We apologize that the description in line 241 is not very accurate. We now changed our description into "Next, we co-transfected mCherry-PIPKI γ and CD3 ζ -GFP into Jurkat cells to visualize CD3 ζ distribution when the clearance of PI(4,5)P $_2$ at the SMAC was impaired" (line 260-262).

Line 245 I find "negative regulation of PI(4,5)P $_2$ " ambiguous. To me, "abnormal PI(4,5)P $_2$ levels and/or distribution" would make sense.

We thank your comment and agree with the suggestion. We have changed our conclusion to "This result indicated that dysfunction of immune synapse response in siNPP5E-transfected cells could be caused by abnormal PI(4,5)P $_2$ levels and/or distribution, similar to the findings in primary cilia" (now in line 264-266).

Line 250 "stained with known" should be "stained with antibodies specific for known"

Fix. Thanks (now in line 271).

Line 258 The sentence ending on line 258 is confusing. Is "(Supplementary Fig. 4c)" the correct figure panel? Should the citation to the figure panel be positioned earlier in the sentence?

Sorry for making the mistake. The correct figure for the sentence ending on line 258 (now in line 280) is the supplementary Fig. 4d. We have corrected it.

Line 259 "these data suggests" should be "these data suggest"

Fix. Thanks (line 280).

Line 272 For clarity, I recommend deleting "the phosphorylation of CD3zeta and ZAP-70 did not appear to be affected by Lck, as"

We agree with your suggestion and have deleted it.

Discussion

Lines 296-299 are difficult to understand, even after consulting reference 37.

Sorry for the unclear statement. We rewrote the sentence to make the description clear (now in line 329-337).

The TCR/CD3 complex is a crucial receptor complex that is involved in T cell activation and phosphoinositides are important in regulating the dynamics of this complex. Studies have shown that

dephosphorylation of PI(4,5)P₂ through overexpressing the plasma membrane-localized inositol polyphosphate-5-phosphatase (Inp54p) in the mouse T-cell hybridoma causes the CD3ε cytoplasmic domain unbinding from the plasma membrane resulting in very modest increase in TCR diffusion rate. In contrast, primary T cells that overexpress PIP5K, an enzyme involved in PI(4,5)P₂ synthesis, increases the amount of PI(4,5)P₂ at the immune synapse, leading to T cell rigidity. It thus delays the recruitment of TCR complex and impairment of proximal TCR signaling. The conclusion of these studies is consistent with the results in our works.

Materials and Methods

Line 357 "The identity of the cell line..." Which cell line(s) does this statement refer to?

The cell lines that we mentioned in the material and methods are Jurkat and Raji cells (now in line 404).

Lines 370 - 374 Please clarify what the described reagents are. Are they plasmids?

These genes indicate CD3ζ and Lck (now in line 420).

Line 401 What data relates to the "Pan T cell isolation kit II mouse"?

We used the pan T cell isolation kit (130-095-130, Miltenyi Biotec) to purify T cells from mice.

Line 484 "Person's" should be "Pearson's"

Fix it. Thanks (now in line 584).

Line 505 "accessed" should be "assessed"

Fixed. Thanks (now in line 609).

Figure Legends

Line 746 Figure 1 legend mentions arrows, but there are no arrows on the images.

Fix it. Thanks

Line 764 and others The legend does not need to say "ns, not significant" when a statistically significant difference is shown.

We agree with the comment and have removed them in the legends.

Line 770 In the 5 min and 10 min images, the bright red signal (INPP5E) is mainly in the CMAC+ cell. Previously, CMAC was used to stain the APC. If the experimental approach in Figure 2h is different, then this should be explained in the Figure Legend.

Thanks for the comment. We realized that the images obtained from epi-fluorescence microscope might not be clear enough to examine the detail localization of INPP5E at the immune synapse. Thus, we used the same experimental protocol and superresolution microscope (3D-SIM) to analyze the

detail localization of INPP5E at immune synapse (Fig 2J). The staining in fig 2J indicated that INPP5E was accumulated at T cells upon APC incubation. Moreover, we used the siINPP5E to confirm that the increase of signal at the immune synapse was INPP5E (Fig. 2b).

Line 783 The word "green" seems to be an error here

Fix it. Thanks (now in line 922).

Line 827 Figure 5c legend mentions arrows, but there are no arrows on the images.

Fix it. Thanks.

Lines 856 - 860 It may be informative to test whether INPP5E-specific siRNA affects IL-2 secretion during "weak" stimulation, "strong" stimulation, or both. Were different concentrations of anti-CD3/CD28 and/or SEE tested? The Figure Legend also does not state how many technical replicates were performed in each experiment.

Thanks for the comment.

(1) We used different concentrations of anti-CD3/CD28 (0.1, 0.5, and 1ug/ml) for T cell activation. Our results showed that knockdown INPP5E attenuated IL-2 secretion in response to anti-CD3/CD28 stimulation in a dose-dependent manner. The data indicate a role for INPP5E in regulating cytokine production (Fig. 6g, line 1091)).

(2) For the ELISA assay, we did three biological replicated for each designed experiment. For every condition, we performed three replicates. We now included the information in the figure legends. "The results were an average of three independent assays. Each assay has three technical replicates and was normalized to the amount of IL-2 secreted by activated Ctrl cells". (now in line 1008-1010)

Supp Fig 1a In the lower panels, what do the arrows point to?

Sorry for the unclear statement. The arrows point to INPP5E signals at centriole (colocalized with CEP164).

Supp Fig 1c Do the upper and lower panels show cells analysed in the absence and presence of SEE, respectively? If so, it should be explained in the figure legend and annotated on the figure.

Fixed. We now add "-SEE" and "+SEE" in the figure legend and figures. (now in line 1022-1023).

Supp Fig 3b Should the lower panels be labelled "Input"?

Fix it. Thanks.

Reviewer #2 (Remarks to the Author):

In this study Chiu and colleagues identify for the first time a role for the ciliary protein INPP5E in the formation of the immune synapse. Using several experimental approaches, the authors show that this ciliary protein localizes to the IS upon superantigen stimulation. Elegant studies showed that downregulation of INPP5E by siRNA impairs CD3Zeta and Zap70 localization to the immune synapse and then clearance of PIP from the IS is compromised leading to less activation of CD3Zeta but not to Lck which localization to the IS is not compromised. The authors performed extended experimentation using complementary techniques to show the importance of INPP5E in the immune synapse, which at least in their experimental systems seem of clear importance in the T cell APC interaction. The findings are novel and relevant in the immunology field. Yet the models used to test these are a bit limited. First, no experiments in primary T cells have been performed to ensure that this is not an artefact resulted from using a transformed cell line, second, most of the evidence comes from very strong CR interactions driven by CD3 antibodies and superantigens that may not represent most of the antigen specific interactions. Yet the evidence provided regarding INPP5E participation in the IS. The presentation of the data is clear, the text is well written and the discussion very thorough. The statistical analyses are correct and all the data regarding experimental repetitions and number of cells analysed is provided.

Major comments

- All the study is performed on the Jurkatt Cell line which is a very good model but still a transformed cell line. The authors should compare the expression levels of INPP5E in normal T cells from several donors to the Jurkatt cell line and at least show that in primary T cells undergoing an immune synapse INPP5E is also directed to the IS.

We thank the reviewer for the precious comments. We agree with the comments and performed additional experiments to validate our findings on primary T cells.

First, we compared the INPP5E levels in T cells from multiple healthy donors to the Jurkat cells (Supplementary Fig. 2c, line 1094). Our results showed that the INPP5E expression levels in primary T cells are comparable to the Jurkat cell line.

We also purified mouse pan T cells and induced immune synapse formation by anti-CD3/CD28 coated beads. Our results showed that INPP5E signals were also enriched at T cells-beads conjugation sites (Supplementary Fig. 2d, line 1094). It indicates that INPP5E is directed to the immune synapse in primary T cells undergoing an immune synapse.

Moreover, we used mouse primary T cells to confirm the importance of INPP5E in immune synapse function. We analyzed *Il-2* mRNA level and found that *Il-2* mRNA level in *shInpp5e*-transduced pan T cells were reduced in compared with that in the control pan T cells. The results have been added to Fig. 6h (line 1091).

-There is little evidence in the overall effect on T cell activation. Please provide data regarding IL-2 production as IL-2 actual concentration and do not normalize the data. Please provide some evidence on early T cell signaling events (NFAT translocation, NFkB and/or AP-1 signaling) and other T cell activation markers (could be CD137, CD69, CD25,..). Alvaro Teijeira, 20.09.22, Immunology and immunotherapy department, CIMA, universidad de NAvarra. Spain.

Thanks for the critical comment. We agree with the comment and performed experiments to check the overall effect of INPP5E on T cell activation (Fig. 6d-6h, also see below; now in line 301-319).

To further examine the influence of INPP5E in early T cell signaling events, we analyzed NF- κ B activity by monitoring its nuclear translocation⁵. The control and siINPP5E Jurkat cells were activated with anti-CD3/CD28 to stimulate the TCR response. The siINPP5E cells showed reduced NF- κ B nuclear translocation compared to control cells, suggesting the importance of INPP5E in early T cell signaling events (Fig. 6d). CD40L is the protein involved in the coordination of immune responses, which is expressed on the surface of activated T cells^{52,53}. In compared to the control cells, the percentage of CD40L-expressing cells was significantly reduced in siINPP5E Jurkat cells, further demonstrating the importance of INPP5E in coordinating immune responses (Fig. 6e). To determine the impact of INPP5E depletion on T cell effector functions, we measured the surface expression of CD25 and the secretion amount of IL-2 in control and siINPP5E Jurkat cells after T cell activation. While INPP5E knockdown only slightly reduced CD25 expression, IL-2 secretion was significantly attenuated in siINPP5E cells compared to control cells, regardless of whether the cells were activated with superantigen-loaded Raji cells or anti-CD3/CD28 (Fig. 6f-g; Supplementary Fig. 5b). We also used primary T cells confirm the importance of INPP5E at the immune synapse. We measured the mRNA level of *Il-2* in shCtrl and sh*Inpp5e* transduced mouse pan T cells. The *Il-2* mRNA level in sh*Inpp5e*-transduced pan T cells was significantly reduced compared to shCtrl-transduced Pan T cells, confirming the essential role of INPP5E in T cell activation (Fig 6h). Together, these data confirm that INPP5E is required for efficient T cell activation through regulation of CD3 ζ and ZAP-70 phosphorylation.

The actual concentration of IL-2 are provided in Fig. 6f and 6g.

Reviewer #3 (Remarks to the Author):

In this manuscript Chiu et al investigate the distribution of cilia-associated proteins in resting and superantigen (SEE)-activated T cells, finding that all the proteins investigated showed a redistribution toward the immunological synapse (IS). The rest of the manuscript focusses on INPP5E, which they show accumulates at the IS of SEE-activated and anti-CD3epsilon/anti-CD28-activated T cells. Using truncation mutants the authors suggest that N-terminal proline-rich domain is responsible for accumulation of INPP5E at the IS. Using immunoprecipitation the authors also suggest there is a direct interaction between INPP5E and CD3zeta, Lck and ZAP70. In contrast to IS accumulation, the C-terminal domain appears to be more important for the proposed interaction with CD3zeta. Knockdown of INPP5E resulted in reduced accumulation of CD3zeta at the IS and defective clearance of PI(4,5)P2 from the center of the synapse, despite normal clearance of actin from the center of the synapse.

INPP5E knockdown also reduced the level of phosphorylated CD3zeta, and ZAP70 in response to TCR and CD28 antibody cross-linking, but did not affect phosphorylated PLCgamma1 or Lck (on the activating Y394 residue), suggesting an effect on proximal signalling efficiency.

Overall, I found the work to be interesting, with many well performed experiments and intriguing observations. My one regret after reading this is that despite the breadth of the results the mechanism underpinning and linking the observations is difficult to glean, although the effect on CD3zeta (and by extension TCR) accumulation and clearance of PI(4,5)P2 from the IS were particularly striking. I would also advise some caution about extending conclusions made about super-antigen stimulated T cells to T cells stimulated conventionally through TCR-agonist peptide MHC interactions. I would also strongly

advise caution about extending conclusions about phosphoinositide biology in Jurkats as this cell line has a documented deficiency in PTEN resulting in constitutive membrane localisation and activity of ITK (doi: 10.1128/MCB.20.18.6945-6957.2000.). Some discussion of how this may impact the translation of findings from this study to primary cells should be made in the Discussion section.

(1) We really thank the reviewer for the comments and suggestions. In our revised manuscript, we changed our wording to avoid making any conclusion in very strong statement without enough evidence throughout the paper.

(2) We understand that the lack of PTEN phosphatase in Jurkat cells can be a strong caveat, especially in studying phosphoinositide metabolism. To strengthen the conclusion regarding the role of INPP5E in immune synapse function, we examined the influence of INPP5E in *Il-2* expression in mouse primary pan T cells. Similar to the results in Jurkat cells, *Il-2* mRNA level was significantly reduced in *Inpp5e*-deficient pan T cells compared to control cells (Fig. 6h). It confirmed that INPP5E attenuated T cell activation both *in vitro* and *in vivo*. We have also discussed the results obtained from primary T cells in our results and discussion section (now in line 382-388.)

I have the following specific comments:

- Fig 2D: CD3zeta and INPP5E do not look colocalised, despite results from pull-downs suggesting they should directly interact. There is also no obvious difference in the accumulation of CD3zeta in the synapse of cells with INPP5E knocked down. Please comment on these discrepancies with other results.

Thanks for the critical comment. We performed the quantification the mean fluorescence intensity of CD3zeta at the immune synapse. Our results showed that intensity of CD3zeta significantly reduced in *siINPP5E* cells compared to the *siCtrl* cells (Fig. 2e). It indicates that INPP5E depletion also reduces the accumulation of CD3zeta at the immune synapse. We apologize that we didn't choose the represented images in the first submitted manuscript. In our revised manuscript, we chose the represented images in Fig. 2d. We also included the quantification of CD3zeta in the revised Fig. 2e.

- Despite good INPP5E knockdown shown in the western blot (Fig 2a) the staining intensity appears similar, with the exception that staining in the IS is abrogated specifically (Fig 2b and Fig 4c). This seems to indicate that there is some non-specific binding of the antibody in the cells, which doesn't affect the conclusions but could at least be commented on briefly when describing Fig 2.

Thanks for the comment and kind suggestion. We agree with the suggestion. In the revised manuscript, we mentioned it when we described Fig. 2 (also see below).

*"Under SEE treatment, although non-specific binding of antibody was found in both T cells and APCs, the polarization of INPP5E toward the immune synapse was significantly reduced in *siINPP5E* cells" (now in line 136-138).*

- Fig 2F: The CD3epsilon staining looks strangely cytoplasmic in -SEE, this doesn't look right at all, it should look membrane bound with some in the ER and Golgi as they traffick to the surface.

We believe that the difference of CD3epsilon staining pattern is due the difference of staining protocol. When we used PFA (4% PFA in PBS, 10 min at room temperature) to fix cells, CD3epsilon signals were

clearly visible at the ER and Golgi (see below). However, this method caused very high staining background of INPP5E staining in our case. To overcome this issue, we used 2% PFA and 0.1% TritonX-100 in PTEM buffer for 10 min at room temperature followed by ice-cold MeOH for another 10 min for fixing and permeabilizing cells (described in Material and Methods). Although this protocol allowed us for clear visualization of INPP5E signaling at the immune synapse, it caused a reduction in CD3e signals at the ER and Golgi.

LCK / CD3e (with 4% PFA and TX100 perm)

- Fig 3D: There is a lot of information missing in the Fig 3 legend. What does each dot represent in the top panel of Fig 3D? Does each dot represent a single cell? What about the bottom panel? The average of values from the two experiments? If so, how did you do statistics on two values? What statistical analysis was used here and what do the symbols indicate?

Sorry for missing the information in Fig. 3 legend. We have added the detail information in the legend of Fig. 3 (now in line 919-933).

Fig. 3: The proline-rich domain is required for efficient INPP5E recruitment at the immune synapse. **a** Diagrams of INPP5E truncations. Proline-rich domain (PRD), inositol polyphosphate phosphatase catalytic (IPPC) domain, CaaX motif are marked in yellow, blue, and purple, respectively. The CTS indicates the ciliary targeting sequence. **b** Jurkat cells were transfected with different truncations of Flag-INPP5E. The expression levels of Flag-INPP5E truncations were examined by western blots. α -tubulin was used as control for equal loading. **c** Immunostaining of Flag-INPP5E truncations in conjugates of Jurkat T cells and CMAC-labeled SEE-pulsed APCs. Cells were co-stained with anti-CD3e as an immune synapse marker. Scale bar: 10 μ m. **d** Quantification of the Flag-INPP5E recruitment index (upper) and the percentage of conjugates with Flag-INPP5E truncations at the immune synapse (lower) is shown. The recruitment of Flag-INPP5E truncations to the immune synapse was shown by the scatter plot graph. 30-50 conjugates were quantified for each INPP5E mutant. Images are representative of two experiments. Error bars indicate mean \pm SD. One-way ANOVA analysis. *P \leq 0.05. **P \leq 0.005. ****P < 0.0001. **e** Summary of INPP5E truncations in the localization of primary cilia and immune synapse (IS).

-Fig. 4G: The example en face inset is very difficult to see, this should be made larger. How many cells are the example live cell images indicative of? Was it only performed once?

(1) Thanks for the comments. We enlarged the images of en faces (Supplementary Fig. 3d and Fig. 5d) for better visualization.

(2) For live cell imaging, the results were from 3-5 cells for each condition from two independent experiments. We have added the information to the legend in Supplementary Fig. 3d and Fig. 5d.

- lines 306-307: I find it difficult to understand what is meant by "actively and passively" here. I would suggest "...directly via enzymatic activity, or indirectly via an alternative mechanism" may more accurately convey the meaning.

We agree with your suggestion and have changed it (now line 343). Thanks.

In particular, I found that many of the conclusions are framed in very strong terms, but the evidence does not fully support some of the statements made. For example:

- Fig 3C: Accumulation at the IS looks very similar in the examples given. From the quantitation it seems there is still significant accumulation without the proline-rich domain, despite it being stated that this domain is "responsible" as is stated for example in the title of the Figure.

(1) We really appreciate your suggestion. We changed our title in Fig. 3 to "The proline-rich domain is required for efficient INPP5E recruitment at the immune synapse".

(2) We went through the paper to avoid making any conclusion in very strong statement without enough evidence.

- The final section title states that "INPP5E is essential for proximal TCR signalling" (also lines 283-284) but the data show a ~40% reduction in pCD3zeta, a non-significant reduction of pPLCg and a ~30% reduction in IL-2 with knockdown of INPP5E. This indicates that INPP5E may play a role in efficient proximal signalling, but is a long way from demonstrating it is essential.

We appreciate your comment. We have changed our statement in the paper into "INPP5E is required for efficient proximal TCR signaling and effector functions" (now in line 283) and the conclusion for this paragraph "Together, these data indicate that INPP5E is required for efficient T cell activation through regulation of CD3 ζ and ZAP-70 phosphorylation (now in line 318-319).

- lines 296-299: This is not accurate, it is not that TCR diffusion "requires" PI(4,5)P₂ dephosphorylation but ref 37 demonstrates that this releases CD3epsilon chains from the membrane resulting in very modest increases in TCR diffusion rates (Fig 3b in ref 37).

Thanks for the comment. We have corrected it (now in line 330-334).

Studies have shown that dephosphorylation of PI(4,5)P₂ through overexpressing the plasma membrane-localized inositol polyphosphate-5-phosphatase (Inp54p) in the mouse T-cell hybridoma causes the CD3 ϵ cytoplasmic domain unbinding from the plasma membrane, resulting in very modest increase in TCR diffusion rate.

Reviewers' comments:

Reviewer #1 (Remarks to the Author):

Overview

The manuscript by Chiu et al. contains interesting and important data. This reviewer is convinced the cilia-associated protein, INPP5E, is recruited to the immune synapse (IS) in Jurkat T cells and primary mouse T cells during activation. Furthermore, this reviewer is convinced that INPP5E silencing perturbs IS formation and T cell activation in Jurkat T cells. This is important because a necessary role for INPP5E in T cell activation has not been demonstrated before, to my knowledge, even though such a role is plausible, given the evidence that cilia-associated proteins and the centrosome have essential roles in IS formation and function.

Major concerns

Despite these positive features, the manuscript is not yet acceptable for publication. To understand the current manuscript, the reader either needs prior knowledge of the field or they need to consult literature not cited in the current manuscript. For a start, the Introduction does not adequately describe the substantial body of published data indicating that ciliary proteins play a role in T cells during centrosome translocation/polarization to the IS during T cell activation. The authors may find authoritative comments on this topic in Douanne et al., *J Cell Biol* 2021 (PMID: 33956049) and Cassioli et al., *J Cell Sci* 2021 (PMID: 34251457). Nevertheless, a role for INPP5E in the IS has not yet been described, so the current manuscript can contribute to the field.

The current manuscript considers diffusion of proteins through the plasma membrane into the IS. However, it overlooks evidence that vesicular transport delivers signaling proteins, including CD3, to the IS during T cell activation (for example, see Soares et al., *J Exp Med* 2013, PMID: 24101378). The data of Soares et al. is of special interest because Lck delivery to the IS occurs independently of the mechanism that delivers LAT and CD3zeta to the IS. Data in the current manuscript are consistent with the idea that INPP5E plays a role in the vesicular transport mechanism that delivers LAT and CD3zeta to the IS. This would support and extend the findings of Jeong et al., *Cell Mol Immunol* 2023 (PMID: 37029318), who showed the intraflagellar transport protein, IFT20, is required for CD3 accumulation in the cSMAC, potentially by mediating transport of CD3 from endosomes to the cSMAC.

(While not essential for publication, data examining the centrosome location in SEE-activated siINPP5E Jurkat T cells would enhance the current manuscript. INPP5E may be required for centrosome docking underneath the IS within the initial 2 minutes of T cell activation. INPP5E might also or alternatively be required for subsequent events, based on its relatively late recruitment to the IS in the time course experiment in Figure 2J.)

The Introduction is also insufficient in its explanation of phosphoinositides. In T cells, the formation and function of the IS involves the depletion and generation of specific phosphoinositide species in a manner that is tightly regulated in space and time by lipases, kinases, and phosphatases. Stating INPP5E is "an enzyme essential for maintaining phosphoinositide levels" is not informative enough. The phosphoinositide substrates and products of INPP5E should be explained. To help the reader understand relationships between the relevant phosphoinositides, enzymes, and fluorescent probes, the current manuscript would be improved by adding a diagram like Figure S1A of Gawden-Bone et al., *Immunity* 2018 (PMID: 30217409), but including PI(3,4,5)P3 as a second substrate of INPP5E. Consider highlighting PI(4,5)P2 (phosphatidylinositol 4,5-bisphosphate) in the Introduction because: (i) PI(4,5)P2 is a substrate of INPP5E; (ii) IS formation involves PI(4,5)P2 clearance from the cSMAC; (iii) PI(4,5)P2 binds to actin-binding proteins; and (iv) the distributions of PI(4,5)P2 and actin in the IS are closely correlated in space and time (Gawden-Bone et al., *Immunity*).

Effects of INPP5E silencing on F-actin clearance are important to clarify. As the current view is that PI(4,5)P2 clearance from the cSMAC is coupled with actin clearance from the cSMAC (Gawden-Bone et al., Immunity), based on Figure 5C we would expect siINPP5E cells to have incomplete clearance of both PI(4,5)P2 and F-actin from the cSMAC. This is supported by Supplementary Figure 4A and B, which indicate that siINPP5E cells have incomplete clearance of F-actin from the cSMAC in conjugates of Jurkat T cells and SEE-pulsed APCs. However, Supplementary Figure 4D indicates that siINPP5E cells have normal clearance of F-actin from the cSMAC after spreading on an anti-CD3/28-coated coverslip. The text puts emphasis on the finding in Supplementary Figure 4D, which seems inappropriate.

Minor concerns

There are many mistakes that must be corrected. Some of these are listed below.

Line 37 – “antibody capping” is an unusual term; “antibody-mediated crosslinking of TCR complexes” is clearer.

Line 46 – Please rephrase “complex (pMHC) complex”. Consider “complex (pMHC) ligand”.

Line 47 – Please replace “MHC” with “pMHC”.

Line 47 – Please replace “results in” with “initiates”. Building a functional immune synapse requires more than TCR binding to pMHC.

Line 48 – Please rephrase “...engagement of TCR and CD28 relocates to the central...” to clarify what relocates to the cSMAC. Try “TCR and CD28 accumulate in the central supramolecular activating complex (cSMAC).”

Lines 49-51 – Please consider deleting “that strengthen the binding affinity to ICAM-1 and VCAM-1”.

Line 51 – “F-actin spreads over the APC surface...” is confusing because the focus of this paper is on events that occur in T cells.

Line 56 – Please replace “activations” with “activation”.

Lines 62-79 – This paragraph refers to “MTOC” and “centrosome”. If both terms must be used, please explain that the centrosome is the MTOC in T cells.

Line 103 – Please delete “A group of”

Line 148 – Please replace “T cells” with “Jurkat T cells”.

Lines 219-220 – When describing ectopic expression of CD3 ζ -GFP, the statement “...GFP signals, which reflected the localization of endogenous CD3 ζ ” seems inappropriate. This raises a question. In resting Jurkat T cells, why is ectopically expressed CD3 ζ -GFP confined to the plasma membrane (Fig. 4F) whereas most endogenous CD3 ζ is inside the cell (Fig 1A-C)?

Lines 305-306 state “CD40L is the protein involved in the coordination of immune responses, which is expressed on the surface of activated T cells.” The grammar is incorrect. An adjectival clause should be placed immediately after the noun or pronoun it describes. This error also occurs on lines 85, 87, 259, 377,

Line 325 – PIP3 is mentioned. What data give in the manuscript give insight into PIP3?

Line 426 – please state whether the Flag tag is at the N- or C-terminus of the INPP5E protein and its truncation variants.

Lines 909 – 910 The figure legend description of Fig 2 panel (i) seems to have swapped the upper and lower summary graphs.

Line 943 – Shouldn't the figure legend state that the Jurkat cells were transfected with a plasmid that encodes siRNA, either control or INPP5E, in Figure 4D?

Line 968 – A paired t-test is inappropriate. An unpaired t-test must be used instead.

Line 1021 – The cell-cycle phases in the Figure Legend do not match those depicted on Supplementary Figure 1B.

fig 2 – panel "g" is missing a label.

fig 5 panel B – the y-axis should start at 0 %, not 40%.

Supp Figure 3D – The label in red font indicates "INPP5E" instead of "F-actin".

Reviewer #2 (Remarks to the Author):

The authors have answered all my comments.

Reviewer #3 (Remarks to the Author):

Thank you for taking on board my comments and for making changes that have improved the manuscript. I have attached a response to your rebuttal with my comments highlighted in red text. My only remaining concern is with the use of "in vivo" to describe the primary T cell work. I outline this in more detail in the attached file.

Reviewer #3 (Remarks to the Author):

In this manuscript Chiu et al investigate the distribution of cilia-associated proteins in resting and superantigen (SEE)-activated T cells, finding that all the proteins investigated showed a redistribution toward the immunological synapse (IS). The rest of the manuscript focusses on INPP5E, which they show accumulates at the IS of SEE-activated and anti-CD3epsilon/anti-CD28-activated T cells. Using truncation mutants the authors suggest that N-terminal proline-rich domain is responsible for accumulation of INPP5E at the IS. Using immunoprecipitation the authors also suggest there is a direct interaction between INPP5E and CD3zeta, Lck and ZAP70. In contrast to IS accumulation, the C-terminal domain appears to be more important for the proposed interaction with CD3zeta. Knockdown of INPP5E resulted in reduced accumulation of CD3zeta at the IS and defective clearance of PI(4,5)P2 from the center of the synapse, despite normal clearance of actin from the center of the synapse.

INPP5E knockdown also reduced the level of phosphorylated CD3zeta, and ZAP70 in response to TCR and CD28 antibody cross-linking, but did not affect phosphorylated PLCgamma1 or Lck (on the activating Y394 residue), suggesting an effect on proximal signalling efficiency.

Overall, I found the work to be interesting, with many well performed experiments and intriguing observations. My one regret after reading this is that despite the breadth of the results the mechanism underpinning and linking the observations is difficult to glean, although the effect on CD3zeta (and by extension TCR) accumulation and clearance of PI(4,5)P2 from the IS were particularly striking. I would also advise some caution about extending conclusions made about super-antigen stimulated T cells to T cells stimulated conventionally through TCR-agonist peptide MHC interactions. I would also strongly advise caution about extending conclusions about phosphoinositide biology in Jurkats as this cell line has a documented deficiency in PTEN resulting in constitutive membrane localisation and activity of ITK (doi: 10.1128/MCB.20.18.6945-6957.2000.). Some discussion of how this may impact the translation of findings from this study to primary cells should be made in the Discussion section.

(1) We really thank the reviewer for the comments and suggestions. In our revised manuscript, we changed our wording to avoid making any conclusion in very strong statement without enough evidence throughout the paper.

Thank you for taking this on board and making changes.

(2) We understand that the lack of PTEN phosphatase in Jurkat cells can be a strong caveat, especially in studying phosphoinositide metabolism. To strengthen the conclusion regarding the role of INPP5E in immune synapse function, we examined the influence of INPP5E in *Il-2* expression in mouse primary pan T cells. Similar to the results in Jurkat cells, *Il-2* mRNA level was significantly reduced in *Inpp5e*-deficient pan T cells compared to control cells (Fig. 6h). It confirmed that INPP5E attenuated T cell activation both *in vitro* and *in vivo*. We have also discussed the results obtained from primary T cells in our results and discussion section (now in line 382-388.)

Lines 386-388: Please reword this sentence. I appreciate the effort in performing this experiment in primary cells and think this significantly strengthens the paper. However, this is not the correct usage of “*in vivo*” — both Jurkat and isolated primary T cell experiments were done *in vitro*. You could simply change the description of the isolated primary T cell work to *ex vivo* and that would be more semantically correct, but I think it would be better to rephrase the sentence to convey

that the results confirm that INPP5E plays a role in T cell activation and support the conclusion that functions of INPP5E in Jurkats are not influenced by other alterations in phosphoinositide signalling. It is of course up to you.

I have the following specific comments:

- Fig 2D: CD3zeta and INPP5E do not look colocalised, despite results from pull-downs suggesting they should directly interact. There is also no obvious difference in the accumulation of CD3zeta in the synapse of cells with INPP5E knocked down. Please comment on these discrepancies with other results.

Thanks for the critical comment. We performed the quantification the mean fluorescence intensity of CD3zeta at the immune synapse. Our results showed that intensity of CD3zeta significantly reduced in siINPP5E cells compared to the siCtrl cells (Fig. 2e). It indicates that INPP5E depletion also reduces the accumulation of CD3zeta at the immune synapse. We apologize that we didn't choose the represented images in the first submitted manuscript. In our revised manuscript, we chose the represented images in Fig. 2d. We also included the quantification of CD3zeta in the revised Fig. 2e.

This looks better and the quantification is a good addition.

- Despite good INPP5E knockdown shown in the western blot (Fig 2a) the staining intensity appears similar, with the exception that staining in the IS is abrogated specifically (Fig 2b and Fig 4c). This seems to indicate that there is some non-specific binding of the antibody in the cells, which doesn't affect the conclusions but could at least be commented on briefly when describing Fig 2.

Thanks for the comment and kind suggestion. We agree with the suggestion. In the revised manuscript, we mentioned it when we described Fig. 2 (also see below).

“Under SEE treatment, although non-specific binding of antibody was found in both T cells and APCs, the polarization of INPP5E toward the immune synapse was significantly reduced in siINPP5E cells” (now in line 136-138).

Thank you for making the change, it is clearer now.

- Fig 2F: The CD3epsilon staining looks strangely cytoplasmic in -SEE, this doesn't look right at all, it should look membrane bound with some in the ER and Golgi as they traffick to the surface.

We believe that the difference of CD3epsilon staining pattern is due the difference of staining protocol. When we used PFA (4% PFA in PBS, 10 min at room temperature) to fix cells, CD3epsilon signals were clearly visible at the ER and Golgi (see below). However, this method caused very high staining background of INPP5E staining in our case. To overcome this issue, we used 2% PFA and 0.1% TritonX-100 in PTEM buffer for 10 min at room temperature followed by ice-cold MeOH for another 10 min for fixing and permeabilizing cells (described in Material and Methods). Although this protocol allowed us for clear visualization of INPP5E signaling at the immune synapse, it caused a reduction in CD3e signals at the ER and Golgi.

LCK / CD3 ϵ (with 4% PFA and TX100 perm)

Thank you for the clarification.

- Fig 3D: There is a lot of information missing in the Fig 3 legend. What does each dot represent in the top panel of Fig 3D? Does each dot represent a single cell? What about the bottom panel? The average of values from the two experiments? If so, how did you do statistics on two values? What statistical analysis was used here and what do the symbols indicate?

Sorry for missing the information in Fig. 3 legend. We have added the detail information in the legend of Fig. 3 (now in line 919-933).

Fig. 3: The proline-rich domain is required for efficient INPP5E recruitment at the immune synapse. **a** Diagrams of INPP5E truncations. Proline-rich domain (PRD), inositol polyphosphate phosphatase catalytic (IPPC) domain, CaaX motif are marked in yellow, blue, and purple, respectively. The CTS indicates the ciliary targeting sequence. **b** Jurkat cells were transfected with different truncations of Flag-INPP5E. The expression levels of Flag-INPP5E truncations were examined by western blots. α -tubulin was used as control for equal loading. **c** Immunostaining of Flag-INPP5E truncations in conjugates of Jurkat T cells and CMAC-labeled SEE-pulsed APCs. Cells were co-stained with anti-CD3 ϵ as an immune synapse marker. Scale bar: 10 μ m. **d** Quantification of the Flag-INPP5E recruitment index (upper) and the percentage of conjugates with Flag-INPP5E truncations at the immune synapse (lower) is shown. The recruitment of Flag-INPP5E truncations to the immune synapse was shown by the scatter plot graph. 30-50 conjugates were quantified for each INPP5E mutant. Images are representative of two experiments. Error bars indicate mean \pm SD. One-way ANOVA analysis. * $P \leq 0.05$. ** $P \leq 0.005$. **** $P < 0.0001$. **e** Summary of INPP5E truncations in the localization of primary cilia and immune synapse (IS).

Thank you for including this information.

-Fig. 4G: The example en face inset is very difficult to see, this should be made larger. How many cells are the example live cell images indicative of? Was it only performed once?

(1) Thanks for the comments. We enlarged the images of en faces (Supplementary Fig. 3d and Fig. 5d) for better visualization.

(2) For live cell imaging, the results were from 3-5 cells for each condition from two independent experiments. We have added the information to the legend in Supplementary Fig. 3d and Fig. 5d.

Thank you for this change.

- lines 306-307: I find it difficult to understand what is meant by "actively and passively" here. I would suggest "...directly via enzymatic activity, or indirectly via an alternative mechanism" may more accurately convey the meaning.

We agree with your suggestion and have changed it (now line 343). Thanks.

This is clearer now.

In particular, I found that many of the conclusions are framed in very strong terms, but the evidence does not fully support some of the statements made. For example:

- Fig 3C: Accumulation at the IS looks very similar in the examples given. From the quantitation it seems there is still significant accumulation without the proline-rich domain, despite it being stated that this domain is "responsible" as is stated for example in the title of the Figure.

(1) We really appreciate your suggestion. We changed our title in Fig. 3 to "The proline-rich domain is required for efficient INPP5E recruitment at the immune synapse".

(2) We went through the paper to avoid making any conclusion in very strong statement without enough evidence.

The wording of the title and conclusions now more accurately reflects the results.

- The final section title states that "INPP5E is essential for proximal TCR signalling" (also lines 283-284) but the data show a ~40% reduction in pCD3zeta, a non-significant reduction of pPLCg and a ~30% reduction in IL-2 with knockdown of INPP5E. This indicates that INPP5E may play a role in efficient proximal signalling, but is a long way from demonstrating it is essential.

We appreciate your comment. We have changed our statement in the paper into "INPP5E is required for efficient proximal TCR signaling and effector functions" (now in line 283) and the conclusion for this paragraph "Together, these data indicate that INPP5E is required for efficient T cell activation through regulation of CD3 ζ and ZAP-70 phosphorylation (now in line 318-319).

- lines 296-299: This is not accurate, it is not that TCR diffusion "requires" PI(4,5)P₂ dephosphorylation but ref 37 demonstrates that this releases CD3epsilon chains from the membrane resulting in very modest increases in TCR diffusion rates (Fig 3b in ref 37).

Thanks for the comment. We have corrected it (now in line 330-334).

Studies have shown that dephosphorylation of PI(4,5)P₂ through overexpressing the plasma membrane-localized inositol polyphosphate-5-phosphatase (Inp54p) in the mouse T-cell hybridoma causes the CD3 ϵ cytoplasmic domain unbinding from the plasma membrane, resulting in very modest increase in TCR diffusion rate.

Thank you for taking this on board, I appreciate the change. One small comment on this: “very modest increase” was my personal interpretation of the difference in diffusion rate and perhaps feels a little out of place in your context. You are of course free to use your own wording or a different qualifying statement, maybe simply “modest increase”.

Point-by-point response

Reviewer #1 (Remarks to the Author):

Overview

The manuscript by Chiu et al. contains interesting and important data. This reviewer is convinced the cilia-associated protein, INPP5E, is recruited to the immune synapse (IS) in Jurkat T cells and primary mouse T cells during activation. Furthermore, this reviewer is convinced that INPP5E silencing perturbs IS formation and T cell activation in Jurkat T cells. This is important because a necessary role for INPP5E in T cell activation has not been demonstrated before, to my knowledge, even though such a role is plausible, given the evidence that cilia-associated proteins and the centrosome have essential roles in IS formation and function.

Major concerns

Despite these positive features, the manuscript is not yet acceptable for publication. To understand the current manuscript, the reader either needs prior knowledge of the field or they need to consult literature not cited in the current manuscript. For a start, the Introduction does not adequately describe the substantial body of published data indicating that ciliary proteins play a role in T cells during centrosome translocation/polarization to the IS during T cell activation. The authors may find authoritative comments on this topic in Douanne et al., J Cell Biol 2021 (PMID: 33956049) and Cassioli et al., J Cell Sci 2021 (PMID: 34251457). Nevertheless, a role for INPP5E in the IS has not yet been described, so the current manuscript can contribute to the field.

We would like to express our gratitude to the reviewer for recognizing the significance of our research and for providing nice suggestions. We have taken the reviewer's advice and included the recommended references in our manuscript. We have now incorporated the current concept in the process of centrosome polarization to the IS during T cell activation in the revised manuscript.

Introduction (Line68-110)

The centrosome, the major microtubule-organizing center (MTOC) in animal cells, plays a critical role in organizing microtubules and acts as the template for primary cilia formation. The similarities between the immune synapse and primary cilia have been increasingly recognized. First, the centrosome polarization at the immune synapse in cytotoxic T lymphocytes (CTL) and its resemblance to centriole docking during ciliogenesis lead to the immune synapse being referred to as a "frustrated cilium"^{9,10}. It is known that docking the centriole to the membrane during ciliogenesis requires the presence of the centriole distal appendage structure at the centriole. Several proteins have been identified to locate at centriole distal appendages, including CEP164, SCLT1, CEP83, CEP89, FBF1, LRRC45, ANKRD26, and TTBK2^{11–13}. Some of these proteins have also been found to localize at the immune synapse in CTL, and the loss of CEP83 impairs CTL secretion¹⁴. Moreover, both immune synapse and primary cilia formation involve actin reorganization and MTOC polarization to facilitate cellular events. The Bardet–Biedl syndrome complex (BBSome), consisting of eight subunits, cooperates with the IFT-B complex and participates in the trafficking of ciliary cargoes ciliary cargo and MTOC-associated functions^{15,16}. BBSome protein BBS1 also helps polarize the centrosome toward the immune synapse and clears F-actin localized around the immune synapse¹⁷. There are also surprising parallels in the trafficking machinery between primary cilia and immune synapses. Several intraflagellar transport proteins (IFTs),

typically associated with primary cilia, are expressed in hematopoietic cells and are involved in T lymphocyte activation. For instance, IFT20 is known to recycle the TCR/CD3 complex and recruit LAT to the immune synapse, supporting the signaling events occurring at the immune synapse¹⁸⁻²². Finally, both structures exhibit concentrated signal transduction activities. At the immune synapse, signaling molecules accumulate in specific regions like the cSMAC to facilitate efficient T-cell signaling. In the primary cilium, the enrichment of receptors in the ciliary membrane receives and transmits signals from the environment, enabling the primary cilium to function as a central signaling hub within the cell.

INPP5E is an enzyme responsible for converting PI(4,5)P₂ and PIP₃ into PI(4)P and PI(3,4)P₂ by removing the 5-phosphate group from them²³. In primary cilia, PI(4,5)P₂ is exclusively found in the ciliary base, while PI(4)P is localized to the ciliary membrane. This specific distribution of phosphoinositides is crucial for Hedgehog signaling, as INPP5E depletion leads to the recruitment of Hedgehog signaling inhibitors like TULP3 and Gpr161^{24,25}. Interestingly, the immune synapse undergoes changes in membrane composition during centrosome docking, similar to the primary cilium. When T cell receptors engage with APCs, phospholipase C-gamma (PLCγ) is rapidly recruited to the immune synapse, leading to the breakdown of PI(4,5)P₂²⁶. The clearance of PI(4,5)P₂ from the cSMAC is essential for efficient centrosome docking and subsequent granule secretion²⁷. On the other hand, the phosphoinositides, especially PI(4,5)P₂ are reported to play important roles in establishing and maintaining the immune synapse²⁸. It is known that the dynamic regulation of actin polymerization at the immune synapse facilitates the delivery of activating signals to the T cell that promotes the communication between the T cell and the APCs. Since PI(4,5)P₂ binds to a variety of actin-regulating proteins, it is likely that PI(4,5)P₂ coordinates the assembly and disassembly of actin filaments by binding to these actin-regulating proteins^{29,30}. Considering the role of INPP5E in controlling the levels of PI(4,5)P₂ and the striking similarities between primary cilia and the immune synapse, it is possible that INPP5E might have a role in the immune synapse by regulating the distribution of phosphoinositides.

The current manuscript considers diffusion of proteins through the plasma membrane into the IS. However, it overlooks evidence that vesicular transport delivers signaling proteins, including CD3, to the IS during T cell activation (for example, see Soares et al., J Exp Med 2013, PMID: 24101378). The data of Soares et al. is of special interest because Lck delivery to the IS occurs independently of the mechanism that delivers LAT and CD3zeta to the IS. Data in the current manuscript are consistent with the idea that INPP5E plays a role in the vesicular transport mechanism that delivers LAT and CD3zeta to the IS. This would support and extend the findings of Jeong et al., Cell Mol Immunol 2023 (PMID: 37029318), who showed the intraflagellar transport protein, IFT20, is required for CD3 accumulation in the cSMAC, potentially by mediating transport of CD3 from endosomes to the cSMAC.

We thank the reviewer for this critical comment. We now have included the concepts of vesicle transport between primary cilia and IS in the introduction, and whether vesicular transport can be the explanation of our results in discussion.

Introduction (Line84-87)

Several intraflagellar transport proteins (IFTs), typically associated with primary cilia, are expressed in hematopoietic cells and are involved in T lymphocyte activation. For instance, IFT20 is known to recycle the TCR/CD3 complex and recruit LAT to the immune synapse, supporting the signaling events occurring at the immune synapse¹⁸⁻²².

Discussion (Line 359-374)

We observe higher levels of PI(4,5)P₂ at the center of the synapse in *siINPP5E* Jurkat cells, along with reduced recruitment of reconstituted CD3ζ-GFP to the immune synapse. This suggests that INPP5E may directly regulate PI(4,5)P₂ levels through its enzymatic activity or indirectly by interacting with other signaling molecules, affecting the diffusion of CD3ζ at the plasma membrane. Another possibility is that INPP5E regulates vesicular transport, which delivers signaling proteins like CD3ζ, to the immune synapse during T cell activation, thereby influencing PI(4,5)P₂ levels. This hypothesis finds support in a published study, which suggests that the delivery of LAT and CD3ζ to the immune synapse happens through a mechanism independent of the one that transports Lck⁵¹. Additionally, the intraflagellar transport protein IFT20 has been shown to regulate CD3/LAT trafficking and is crucial for CD3 accumulation in the central cSMAC^{20,22,69}. Our current findings align with these studies, as we demonstrated that INPP5E affects the recruitment of CD3ζ but not Lck to the immune synapse. Since PI(4,5)P₂ is known to be crucial for regulating actin dynamics and vesicle trafficking at the immune synapse formation by binding to actin-regulating proteins, it would be interesting to explore in future investigation whether INPP5E directly or indirectly influences actin dynamics at synapse²⁸.

(While not essential for publication, data examining the centrosome location in SEE-activated *siINPP5E* Jurkat T cells would enhance the current manuscript. INPP5E may be required for centrosome docking underneath the IS within the initial 2 minutes of T cell activation. INPP5E might also or alternatively be required for subsequent events, based on its relatively late recruitment to the IS in the time course experiment in Figure 2J.)

We thank the reviewer for proposing an interesting experiment, and we believe whether INPP5E is required for centrosome docking and subsequent events will be subject of our future investigations.

The Introduction is also insufficient in its explanation of phosphoinositides. In T cells, the formation and function of the IS involves the depletion and generation of specific phosphoinositide species in a manner that is tightly regulated in space and time by lipases, kinases, and phosphatases. Stating INPP5E is “an enzyme essential for maintaining phosphoinositide levels” is not informative enough. The phosphoinositide substrates and products of INPP5E should be explained. To help the reader understand relationships between the relevant phosphoinositides, enzymes, and fluorescent probes, the current manuscript would be improved by adding a diagram like Figure S1A of Gawden-Bone et al., *Immunity* 2018 (PMID: 30217409), but including PI(3,4,5)P₃ as a second substrate of INPP5E. Consider highlighting PI(4,5)P₂ (phosphatidylinositol 4,5-bisphosphate) in the Introduction because: (i) PI(4,5)P₂ is a substrate of INPP5E; (ii) IS formation involves PI(4,5)P₂ clearance from the cSMAC; (iii) PI(4,5)P₂ binds to actin-binding proteins; and (iv) the distributions of PI(4,5)P₂ and actin in the IS are closely correlated in space and time (Gawden-Bone et al., *Immunity*).

We thank the referee for the suggestion to provide more background information on INPP5E and phosphoinositides. In the revised manuscript, we provided a comprehensive overview of the role of PI(4,5)P₂ in the immune synapse. We also included a diagram in Supplementary Fig4a that illustrates the substrates of INPP5E and the relative probes used to study phosphoinositide distribution.

Introduction (Line 93-110)

INPP5E is an enzyme responsible for converting PI(4,5)P₂ and PIP₃ into PI(4)P and PI(3,4)P₂ by removing the 5-phosphate group from them²³. In primary cilia, PI(4,5)P₂ is exclusively found in the ciliary base, while PI(4)P is localized to the ciliary membrane. This specific distribution of phosphoinositides is crucial for Hedgehog signaling, as INPP5E depletion leads to the recruitment of Hedgehog signaling inhibitors like TULP3 and Gpr161^{24,25}. Interestingly, the immune synapse undergoes changes in membrane composition during centrosome docking, similar to the primary cilium. When T cell receptors engage with APCs, phospholipase C-gamma (PLC γ) is rapidly recruited to the immune synapse, leading to the breakdown of PI(4,5)P₂²⁶. The clearance of PI(4,5)P₂ from the cSMAC is essential for efficient centrosome docking and subsequent granule secretion²⁷. On the other hand, the phosphoinositides, especially PI(4,5)P₂ are reported to play important roles in establishing and maintaining the immune synapse²⁸. It is known that the dynamic regulation of actin polymerization at the immune synapse facilitates the delivery of activating signals to the T cell that promotes the communication between the T cell and the APCs. Since PI(4,5)P₂ binds to a variety of actin-regulating proteins, it is likely that PI(4,5)P₂ coordinates the assembly and disassembly of actin filaments by binding to these actin-regulating proteins^{29,30}. Considering the role of INPP5E in controlling the levels of PI(4,5)P₂ and the striking similarities between primary cilia and the immune synapse, it is possible that INPP5E might have a role in the immune synapse by regulating the distribution of phosphoinositides.

Sup Fig 4a.

a

Phosphoinositide	Common Probe
PI(4)P	P4M-SidM
PI(4,5)P ₂	PH-PLC δ / Tubby
PIP ₃	PH-AKT
PI(3,4)P ₂	NES-Bam32-PH

Effects of INPP5E silencing on F-actin clearance are important to clarify. As the current view is that PI(4,5)P₂ clearance from the cSMAC is coupled with actin clearance from the cSMAC (Gawden-Bone et al., Immunity), based on Figure 5C we would expect siINPP5E cells to have incomplete clearance of both PI(4,5)P₂ and F-actin from the cSMAC. This is supported by Supplementary Figure 4A and B, which indicate that siINPP5E cells have incomplete clearance of F-actin from the cSMAC in conjugates of Jurkat T cells and SEE-pulsed APCs. However, Supplementary Figure 4D indicates that siINPP5E cells have normal clearance of F-actin from the cSMAC after spreading on an anti-CD3/28-coated coverslip. The text puts emphasis on the finding in Supplementary Figure 4D, which seems inappropriate.

We appreciate the reviewer for the insightful comments and have made several modifications to the revised manuscript. First, we quantified F-actin clearance at the immune synapse. In Jurkat-APC conjugated cells, F-actin staining showed that F-actin clearance was indeed impaired in *siINPP5E* cells, as evidenced by a clearance percentage of 53.6%, compared to 73.6% in control cells (Supplementary Figure 4e). This observation aligns with the inhibitory effect of PI(4,5)P₂ inhibition on F-actin clearance. We also quantified the spreading areas of F-actin on anti-CD3/CD28-coated slides using a quantification method described by Kim et al. (PMID 24454796). We found there were no significant alterations between the control and *siINPP5E* cells (Supplementary Fig 4D, currently Supplementary Figure 4f). We think that the difference of F-actin staining may due to the differences of the activation protocols. Thus, we have revised the manuscript to avoid any confusion or overemphasis on the findings in Supplementary Figure 4e (currently Supplementary Figure 4f), ensuring that our description accurately represents the significance of these results to the readers.

Line 296-304

Given that F-actin clearance is impaired during PI(4,5)P₂ inhibition, we examined the potential influence of INPP5E in regulating F-actin clearance at the immune synapse^{27,28}. In Jurkat-APC conjugation experiments, the F-actin staining showed that F-actin clearance at immune synapse was impaired in *siINPP5E* cells, as evidenced by a clearance percentage of 53.6 compared to 73.6% in control cells (Supplementary Figure 4e), while we did not observe significant alterations in INPP5E-depleted cells when quantifying the spreading areas of F-actin on anti-CD3/CD28-coated slides (Supplementary Figure 4f). Thus, we propose that INPP5E directly modulates PI(4,5)P₂ level at the synapse, which in turn affects the recruitment of CD3ζ and F-actin clearance.

Minor concerns

There are many mistakes that must be corrected. Some of these are listed below.

Thanks for the comments. We have revised the manuscript to address the issue.

Line 37 – “antibody capping” is an unusual term; “antibody-mediated crosslinking of TCR complexes” is clearer.

Fixed. Thanks

Line 46 – Please rephrase “complex (pMHC) complex”. Consider “complex (pMHC) ligand”.

Fixed. Thanks

Line 47 – Please replace “MHC” with “pMHC”.

Fixed. Thanks

Line 47 – Please replace “results in” with “initiates”. Building a functional immune synapse requires more than TCR binding to pMHC.

Thanks for the kind suggestion. We have corrected the mistake (line 48)

Line 48 – Please rephrase “...engagement of TCR and CD28 relocates to the central...” to clarify what relocates to the cSMAC. Try “TCR and CD28 accumulate in the central supramolecular activating complex (cSMAC).”

Fixed. Thanks (line 50-52)

The cSMAC, located at the center of the synapse, is the site where the TCR and CD28 molecules accumulate upon conjugation.

Lines 49-51 – Please consider deleting “that strengthen the binding affinity to ICAM-1 and VCAM-1”.

Deleted. Thanks for the suggestion.

Line 51 – “F-actin spreads over the APC surface...” is confusing because the focus of this paper is on events that occur in T cells.

Sorry for the confusing statement. We have rephrased the sentence to make it clear (now in line54-56).

In addition to the pSMAC and cSMAC, there is an outer ring called the dSMAC, enriched with actin filaments and actin-associated proteins, regulating the cytoskeletal reorganization at the immune synapse.

Line 56 – Please replace “activations” with “activation”.

We have made the suggested replacement. Thanks

Lines 62-79 – This paragraph refers to “MTOC” and “centrosome”. If both terms must be used, please explain that the centrosome is the MTOC in T cells.

Thanks for the comment. We added a sentence to explain that the centrosome is also the MTOC (now in line 68-69).

The centrosome, the major microtubule-organizing center (MTOC) in animal cells, plays a critical role in organizing microtubules and acts as the template for primary cilia formation.

Line 103 – Please delete “A group of”

Deleted. Thanks (line 122).

Line 148 – Please replace “T cells” with “Jurkat T cells”.

Fixed. Thanks (line 167).

Lines 219-220 – When describing ectopic expression of CD3ζ-GFP, the statement “...GFP signals, which reflected the localization of endogenous CD3ζ” seems inappropriate. This raises a question. In resting Jurkat T cells, why is ectopically expressed CD3ζ-GFP confined to the

plasma membrane (Fig. 4F) whereas most endogenous CD3 ζ is inside the cell (Fig 1A-C)?

We thank the reviewer for correcting our mistake. We have corrected the sentence (line 234-236).

To further examine how INPP5E regulated spatial and temporal localization of CD3 ζ at the immune synapse during T cell activation, we ectopically expressed CD3 ζ fused with EGFP into control and siINPP5E cells

We also thank you for the comment. We believe that the difference between CD3z-GFP and endogenous CD3z staining pattern is due the difference of staining protocol. For CD3z-GFP experiment (Figure 4f), we used PFA (4% PFA in PBS, 10 min at room temperature) to fix cells in order to keep the signals of F-actin. However, this method caused very high staining background of INPP5E and ciliary protein staining in our cases. To overcome this issue, we used 2% PFA and 0.1% TritonX-100 in PTEM buffer for 10 min at room temperature followed by ice-cold MeOH for another 10 min for fixing and permeabilizing cells (described in Material and Methods, for Figure 1a-c). Although this protocol allowed us for clear visualization of ciliary protein at the immune synapse, it caused a reduction of surface CD3zeta signals.

Lines 305-306 state “CD40L is the protein involved in the coordination of immune responses, which is expressed on the surface of activated T cells.” The grammar is incorrect. An adjectival clause should be placed immediately after the noun or pronoun it describes. This error also occurs on lines 85, 87, 259, 377,

Thanks for the comment. We have changed the adjectival clause accordingly.

Line 325 – PIP3 is mentioned. What data give in the manuscript give insight into PIP3?

Sorry for the error. We have now removed this term.

Line 426 – please state whether the Flag tag is at the N- or C-terminus of the INPP5E protein and its truncation variants.

The tag is at the N-terminus. Fixed. Thanks. (now in line 455)

Lines 909 – 910 The figure legend description of Fig 2 panel (i) seems to have swapped the upper and lower summary graphs.

Thanks for the comment. We also changed the y-axis (now starts for 0) and used the scatter dot plot instead to be consistent with figures in the manuscript.

Line 943 – Shouldn't the figure legend state that the Jurkat cells were transfected with a plasmid that encodes siRNA, either control or INPP5E, in Figure 4D?

Fix. Thanks (now in line 993)

Line 968 – A paired t-test is inappropriate. An unpaired t-test must be used instead.

Fix. The statistics are changed accordingly. Thanks

Line 1021 – The cell-cycle phases in the Figure Legend do not match those depicted on Supplementary Figure 1B

Sorry for the mistake. We have corrected (line 1073).

fig 2 – panel “g” is missing a label.

Fixed. Thanks.

fig 5 panel B – the y-axis should start at 0 %, not 40%.

Fixed. Thanks.

Supp Figure 3D – The label in red font indicates “INPP5E” instead of “F-actin”.

Fixed. Thanks.

Reviewer #2 (Remarks to the Author):

The authors have answered all my comments.

We really appreciate Reviewer #2 for supportive comments.

Reviewer #3 (Remarks to the Author):

Thank you for taking on board my comments and for making changes that have improved the manuscript. I have attached a response to your rebuttal with my comments highlighted in red text. My only remaining concern is with the use of "in vivo" to describe the primary T cell work. I outline this in more detail in the attached file.

We thank you Reviewer #3 for your supportive comments.

Lines 386-388: Please reword this sentence. I appreciate the effort in performing this experiment in primary cells and think this significantly strengthens the paper. However, this is not the correct usage of “in vivo” — both Jurkat and isolated primary T cell experiments were done in vitro. You could simply change the description of the isolated primary T cell work to ex vivo and that would be more semantically correct, but I think it would be better to rephrase the sentence to convey that the results confirm that INPP5E plays a role in T cell activation and support the conclusion that functions of INPP5E in Jurkats are not influenced by other alterations in phosphoinositide signalling. It is of course up to you.

Thanks for the comment. We now change the in vivo into ex vivo (now in line 416).

Line 296-299

Thank you for taking this on board, I appreciate the change. One small comment on this: “very

modest increase” was my personal interpretation of the difference in diffusion rate and perhaps feels a little out of place in your context. You are of course free to use your own wording or a different qualifying statement, maybe simply “modest increase”.

Thanks for the comment. We have changed the wording accordingly (now in line 355).

Reviewers' comments:

Reviewer #1 (Remarks to the Author):

Most of my comments have been addressed satisfactorily by the revisions. However, there are still some concerns, as listed below.

Major comment.

The revised manuscript seems evasive about TCR signaling proteins reaching the immunological synapse (IS) via recycling endosomes, also called vesicular transport. Evidence for this mechanism is published, for example in PMID: 15142526, 19855387 and 2410137. The Introduction should explicitly state that TCR signaling proteins are thought to reach the IS via at least two mechanisms: via lateral diffusion within the plasma membrane and via recycling endosomes actively transported to the IS. The reader should be armed with this information before they assess the Results.

The evasiveness persists in the revised Discussion, which does mention "vesicular transport", but the sentence does not have any citation(s) and contains a distracting comment about "influencing PI(4,5)P2 levels".

Minor comments.

In Supplementary Figure 4A, in the reactions depicted on the top line, the double arrows point in the wrong direction.

To my comment, "fig 5 panel B – the y-axis should start at 0 %, not 40%", the authors replied "Fixed. Thanks."; however, Figure 5B remains unchanged.

Some errors and omissions in the paragraph on lines 58 to 66 are listed:

"Thee phosphorylated..."

"It leads to..." should be replaced with "This leads to..."

"undergo rapidly actin and microtubule cytoskeletons rearrangement" should be rephrased

Line 81 states "...ciliary cargoes ciliary cargo..."

Point-by-point response

Reviewer #1 (Remarks to the Author):

Most of my comments have been addressed satisfactorily by the revisions. However, there are still some concerns, as listed below.

Major comment.

The revised manuscript seems evasive about TCR signaling proteins reaching the immunological synapse (IS) via recycling endosomes, also called vesicular transport. Evidence for this mechanism is published, for example in PMID: 15142526, 19855387 and 2410137. The Introduction should explicitly state that TCR signaling proteins are thought to reach the IS via at least two mechanisms: via lateral diffusion within the plasma membrane and via recycling endosomes actively transported to the IS. The reader should be armed with this information before they assess the Results.

The evasiveness persists in the revised Discussion, which does mention "vesicular transport", but the sentence does not have any citation(s) and contains a distracting comment about "influencing PI(4,5)P₂ levels".

We extend our sincere appreciation to the reviewer for acknowledging the importance of our research and offering valuable suggestions. Following the reviewer's advice, we have included the recommended references in our manuscript. Moreover, we have incorporated the current concept in the TCR recruiting activity in the revised manuscript.

In the introduction (line 56-60)

Various pathways for TCR recruitment to the immune synapse have been investigated, such as passive lateral diffusion within the plasma membrane^{3,4}, vesicular transport through recycling endosome trafficking⁵, and cytoskeleton-mediated active movement⁶. All these routes are important in facilitating T-cell signaling.

Discussion (line 367-373)

Another possibility is that INPP5E regulates vesicular transport, which delivers signaling proteins like CD3 ζ , to the immune synapse during T cell activation⁵. This hypothesis finds supports in published studies, which suggest that the delivery of LAT and CD3 ζ to the immune synapse happens through a mechanism independent of the one that transports Lck⁵⁵. Additionally, the intraflagellar transport protein IFT20 has been shown to regulate CD3/LAT trafficking and is crucial for CD3 accumulation in the central cSMAC^{24,26,73}. Our current findings align with these studies, as we demonstrated that INPP5E affects the recruitment of CD3 ζ but not Lck to the immune synapse. Since PI(4,5)P₂ is known to be crucial for regulating actin dynamics and vesicle trafficking at the immune synapse formation by binding to actin-regulating proteins, it would be interesting to investigate whether INPP5E influences TCR through vesicular trafficking or actin dynamics at the immune synapse³².

Minor comments.

In Supplementary Figure 4A, in the reactions depicted on the top line, the double arrows point in the wrong direction.

Fixed. Thanks.

To my comment, "fig 5 panel B – the y-axis should start at 0 %, not 40%", the authors replied "Fixed. Thanks."; however, Figure 5B remains unchanged.

We are very sorry about this. We have corrected the mistake.

Some errors and omissions in the paragraph on lines 58 to 66 are listed:

"Thee phosphorylated..."

"It leads to..." should be replaced with "This leads to..."

"undergo rapidly actin and microtubule cytoskeletons rearrangement" should be rephrased

Line 81 states "...ciliary cargoes ciliary cargo..."

We fixed these errors accordingly.